# ROYAL SOCIETY
# OPEN SCIENCE

rsos.royalsocietypublishing.org

Subject Areas:
statistics

Keywords:
survival analysis, Kaplan–Meier, heterogeneous distribution, non-parametric, hypothesis test, asymptotic analysis

Author for correspondence:
Joshua C. Chang
e-mail: josh.chang@nih.gov

# Asymptotic convergence in distribution of the area bounded by prevalence-weighted Kaplan–Meier curves using empirical process modelling

Aaron Heuser[2], Minh Huynh[2] and Joshua C. Chang[1]

[1]Epidemiology and Biostatistics Section, Rehabilitation Medicine Department, The National Institutes of Health Clinical Center, Bethesda, MD 20892, USA
[2]Impaq International LLC, Washington, DC 20005, USA

JCC, 0000-0001-9690-9179

The Kaplan–Meier product-limit estimator is a simple and powerful tool in time to event analysis. An extension exists for populations stratified into cohorts where a population survival curve is generated by weighted averaging of cohort-level survival curves. For making population-level comparisons using this statistic, we analyse the statistics of the area between two such weighted survival curves. We derive the large sample behaviour of this statistic based on an empirical process of product-limit estimators. This estimator was used by an interdisciplinary National Institutes of Health–Social Security Administration team in the identification of medical conditions to prioritize for adjudication in disability benefits processing.

## 1. Introduction

Survival analysis addresses the classical statistical problem of determining characteristics of the waiting time until an event, canonically death, from observations of their occurrence sampled from within a population. This problem is not trivial as the expected waiting time is typically dependent on the time-already-waited. For instance, a hundred-year-old can be more certain of surviving to his or her one hundred-and-first birthday than a newborn might reasonably be. However, the comparison may shift in the newborn's favour for the living to 121, particularly in light of medical advances that make survival probabilities

**Figure 1.** Inhomogeneity of survival within populations can result due to at least two reasons. In (*a*), inhomogeneity results from a categorical covariate that influences survival statistics. In (*b*), inhomogeneity results from non-stationarity, where cohorts of individuals are sampled at different times. In this case, the problem of progressive censoring is apparent because later cohorts have not been observed as long.

rsos.royalsocietypublishing.org    R. Soc. open sci. **5**: 180496

non-stationary. Parametric approaches for assembling survival curves are usually not flexible enough to capture this complexity.

One simple approach to this problem was pioneered by the work of Kaplan & Meier [1]. Their product-limit estimator [2–5] is a non-parametric statistic that is used for inferring the survival function for members of a population from observed lifetimes. This method is particularly useful in that it naturally handles the presence of right censoring, where some event times are only partially observed because they fall outside the observation window. It was not, however, designed to account for varying subpopulations that may yield non-homogeneity in overall population survival (figure 1). For instance, in the example given above, subpopulations for survival characteristics may be defined by birth year or entry cohort of a subject in a particular study (figure 1).

Several existing statistical methods address variants of this limitation. A natural approach is to consider the varying subpopulations as defining underlying covariates, thus laying the framework for a proportional hazards model. The assumption of proportional hazards is quite strong. When considering time-dependent statistics (as in the motivational example), it is violated in all but a few specific cases. Likewise, frailty models, first developed by Hougaard (cf. [6]), and extended by Aalen (cf. [7]), assume multivariate event distributions, but also make assumptions on the underlying event distributions and assume proportional hazards.

Other existing methods, such as bivariate survival analysis (cf. [8]), consider the time to observation and the time to event as conditionally independent random times. Underlying these methods is the assumption that upon the time of observation, all individuals will then have a similar event time distribution, thus failing to acknowledge the temporal changes.

These complexities arose in the identification of new disorders to incorporate into the United States Social Security Administration (SSA)'s Compassionate Allowances (CAL) initiative. The CAL initiative seeks to identify candidate medical conditions for fast-tracking in the processing of disability applications. The intent of this initiative is to prioritize applicants who are most likely to die in the time-course of usual case processing so that they may receive benefits while still living.

At its inception, the CAL initiative identified conditions based on the counsel of expert opinion [9]. The SSA in collaboration with the National Institutes of Health (NIH) sought to expand the list of CAL conditions systematically, using a databased approach. Using in part the survival estimator described in this paper, the NIH identified 24 conditions for inclusion into the list of conditions [9].

The methodology used in CAL is related to that of the work of Pepe & Fleming (cf. [10,11]), where a class of weighted Kaplan–Meier statistics is introduced. Though these statistics exhibit the same limitations as in the standard Kaplan–Meier case, it should be noted that [11] introduces the stratified weighted Kaplan–Meier statistic. The statistic presented here is *a priori* quite similar, but instead of a weighting function, includes the empirical prevalence. In doing so, the weight is no longer independent of the event time estimate, and thus requires much different methods of proof.

rsos.royalsocietypublishing.org    R. Soc. open sci. **5**: 180496

We thus consider the overall survival distribution for a population of individuals with sub-populations that exhibit non-homogeneous survival distributions. Through this consideration, a new test statistic, based upon the empirical process of product-limit estimators is developed. Through constructive methods, this test statistic compares survival distributions among the distinct subpopulations, and weights according to distribution of the identified subgroups.

# 2. Statistical method

Suppose $\Gamma^{(1)}$ and $\Gamma^{(2)}$ are disjoint populations of individuals where each individual belongs to exactly one of $d$ distinct cohorts labelled $z \in \mathbb{Z}_d$. For randomly selected individuals $\gamma \in \Gamma^{(i)}$ within population $i$, we desire to understand the statistics of the event time $T^\gamma$ under the assumption that survival is conditional on cohort $z^\gamma$ and population.

One representation of the marginal survival probability for members of population $i$, $\theta_t^{(i)} = \mathbb{P}\{T^\gamma > t \mid \gamma \in \Gamma^{(i)}\}$, is found by conditioning on cohort

$$\theta_t^{(i)} = \sum_{z=1}^{d} \underbrace{\mathbb{P}\{T^\gamma > t \mid z^\gamma = z, \gamma \in \Gamma^{(i)}\}}_{S_{z,t}^{(i)}} \underbrace{\mathbb{P}\{z^\gamma = z \mid \gamma \in \Gamma^{(i)}\}}_{q_z^{(i)}}, \tag{2.1}$$

where $S_{z,t}^{(i)}$ represents the survival function for individuals of cohort $z$ in population $i$, where each individual's cohort membership is known.

We use this representation of the survival probability as motivation to formulate an estimator for the population-average survival functions

$$\hat{\theta}_t^{(i)} = \sum_{z=1}^{d} \hat{q}_z^{(i)} \hat{S}_{z,t}^{(i)}, \tag{2.2}$$

where $\hat{q}_z^{(i)}$ and $\hat{S}_{z,t}^{(i)}$ are estimators of the cohort prevalence and cohort-wise survival, respectively. This weighted Kaplan–Meier method has appeared previously in the literature [12], and has been empirically validated against the pure Kaplan–Meier method [13], where the weighting procedure was found to reduce the bias in the construction of survival curves. The asymptotic convergence of the product-limit estimator and weighted variants is well established [11,14]. We use this survival curve reconstruction method as a base in constructing a new statistic for comparing populations. The focus of this paper is not the properties of this survival estimator but rather the asymptotic convergence of its bounding area and the use of such a quantity for evaluating a null hypothesis.

Our concern is the general situation where random samples of size $n^{(i)}$ are chosen from each of the respective populations. Within these samples, the number of individuals within each cohort, $n_z^{(i)}$, is counted, from which an estimator of the cohort distribution is obtained

$$\hat{q}_z^{(i)} = \frac{n_z^{(i)}}{n^{(i)}}. \tag{2.3}$$

In turn, we assume that the cohort-level survival functions $\hat{S}_{z,t}^{(i)}$ are estimated independently using the product-limit estimator. Note that since the product-limit estimator is not a linear functional of sampled lifetimes, $\hat{\theta}_t^{(i)}$ is distinct from the estimator obtained by applying the product-limit estimator directly on all $n^{(i)}$ samples of population $i$. To prevent confusion, we denote all direct applications of the product-limit estimator using $\hat{S}$ and all instances of weighted sums of product-limit estimators using the Greek letter $\hat{\theta}$.

With these elements in place, we define our test statistic

$$\hat{\Theta} = \sqrt{\frac{n^{(1)}n^{(2)}}{n^{(1)} + n^{(2)}}} \int_0^\tau \mathrm{d}t (\hat{\theta}_t^{(1)} - \hat{\theta}_t^{(2)}), \tag{2.4}$$

where $\tau = \inf\{\tau_z : z \in \mathbb{Z}_d\}$, and $\tau_z$ denotes the time at which cohort $z$ is censored in observations. Note that in the absence of random prevalence this statistic is equivalent to comparison of mean lifetimes between the two populations [10]. We state here the main result of the paper—the large sample behaviour of this statistic within a null-hypothesis statistical testing framework.

**Theorem 2.1.** *Let $C_{z,t}^{(i)}$ denote the probability that a z-type individual has not yet been censored at time $t \geq 0$ (the survival probability relative to the occurrence of censoring), and $q_z^{(i)}$ denote the probability that an individual in*

population $i$ is of cohort $z$, and let $p^{(i)} = n^{(i)}/(n^{(1)} + n^{(2)})$. Suppose that $\theta_t^{(1)} = \theta_t^{(2)}$. Then $\hat{\Theta} \xrightarrow{d} N(0, \sigma^2)$, as $n^{(i)} \to \infty$, with

$$\sigma^2 = \sum_{i=1}^2 (1 - p^{(i)}) \left( \sum_{z=1}^d q_z^{(i)} \phi_z^2 - \left( \sum_{z=1}^d q_z^{(i)} \phi_z \right)^2 \right) - \sum_{z=1}^d \int_0^{\tau_z} \mathrm{d}S_{z,t} W_{z,t} \times \left( \frac{\phi_{z,t}}{S_{z,t}} \right)^2,$$

where for $0 \le t \wedge \tau_z$, where $\tau_z$ is the time at which samples of cohort $z$ are censored, $\phi_{z,t} = \int_t^{\tau_z} \mathrm{d}s\, S_{z,s}$, $\phi_z \equiv \phi_{z,0}$, $S_{z,t}$ is the survival function for the pooled data of cohort $z$, and

$$W_{z,t} = \left( \frac{p^{(1)} C_{z,t-}^{(1)} q_z^{(2)} + p^{(2)} C_{z,t-}^{(2)} q_z^{(1)}}{C_{z,t-}^{(1)} C_{z,t-}^{(2)}} \right).$$

Note that this quantity is well defined since by definition of $\tau_z$, $C_{z,t}^{(z)} > 0$ for all $t \le \tau_z$. The variance $\sigma^2$ may be consistently estimated by

$$\hat{\sigma}^2 = \sum_{i=1}^2 (1 - p^{(i)}) \left( \sum_{z=1}^d \hat{q}_z^{(i)} \hat{\phi}_z^2 - \left( \sum_{z=1}^d \hat{q}_z^{(i)} \hat{\phi}_z \right)^2 \right) - \sum_{z=1}^d \int_0^{\tau_z} \mathrm{d}\hat{S}_{z,t}\, \hat{W}_{z,t} \times \left( \frac{\hat{\phi}_{z,t}}{\hat{S}_{z,t}} \right)^2, \tag{2.5}$$

where for $0 \le t \wedge \tau_z$, $\hat{S}_{z,t}$ is the product-limit estimate of the pooled data for cohort $z$,

$$\hat{\phi}_{z,t} = \int_t^{\tau_z} \mathrm{d}s\hat{S}_{z,s}, \tag{2.6}$$

$\hat{C}_{z,t}^{(i)}$ is the product-limit estimate associated with the event of censoring for cohort $z$ within population $i$, $\hat{\phi}_z \equiv \hat{\phi}_{z,0}$, and

$$\hat{W}_{z,t} = \left( \frac{p^{(1)} \hat{C}_{z,t-}^{(1)} \hat{q}_z^{(2)} + p^{(2)} \hat{C}_{z,t-}^{(2)} \hat{q}_z^{(1)}}{\hat{C}_{z,t-}^{(1)} \hat{C}_{z,t-}^{(2)}} \right). \tag{2.7}$$

Note that this quantity is also well defined since $\hat{C}_{z,t}^{(z)} > 0$ for all $t \le \tau_z$. In appendix A, we provide a proof of theorem 2.1 in an empirical process framework. Note that since survival estimates $\hat{\theta}$ and $\hat{S}$ are step functions, all integrals are exactly computable.

# 3. Numerical investigation

A computational implementation of the test statistic $\hat{\Theta}$ and weighted survival estimators is available in the form of a package for R. This package also contains a class to handle arithmetic involving right-continuous piecewise linear functions. In the appendices, we have provided source code that may be used for installing and invoking this package.

Here, we present a computational investigation of the weighted survival curve estimator and the corresponding test statistic. Using simulations, we investigated the statistical power of $\hat{\Theta}$, contrasted with that of existing non-parametric methods. Using a real dataset, we demonstrate the computation of $\hat{\Theta}$, $\hat{\theta}_t$, and evaluate type I error.

## 3.1. Evaluating statistical power through simulations

Using simulations, we explored the statistical power of the test statistic $\hat{\Theta}$ in a case where populations are difficult to distinguish based purely on mean survival time. As test populations, we examined admixtures of exponential and Weibull distributions for the event time, and compared survival in these mixture populations to survival of a population of purely exponential event times (figure 2). Population 1 consists of individuals having an exponentially distributed lifetime with a mean of $\lambda^{-1} = 4$ years. Population 2 consists of two types of individuals: those who have an exponentially distributed lifetime with a mean of 5 years (type $z = 1$), and those of type $z = 2$ who have a Weibull distributed lifetime with shape parameter $k = 5$ and scale parameter $\lambda = 1$.

Since Population 1 is homogeneous, we only track subpopulations of Population 2—we drop the superscript and denote the proportion of Population 2's members of type 2 by $q_2$. It is most instructive to examine our method in the neighbourhood where both populations have approximately the same expectation value for the event time, which occurs for $q_2 \approx 0.245$. For this reason, we chose values near 0.25 for our simulations.

rsos.royalsocietypublishing.org    R. Soc. open sci. **5**: 180496

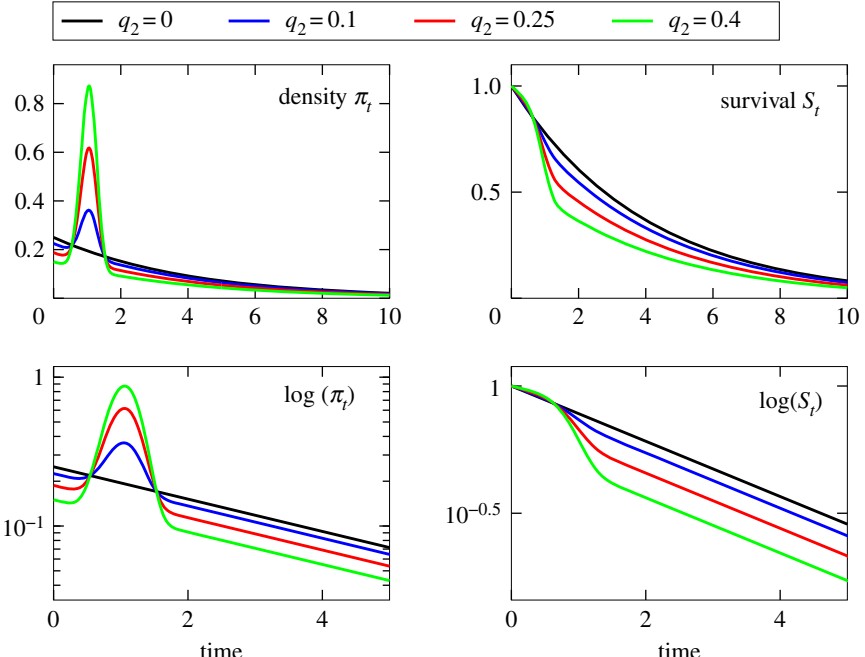

**Figure 2.** Admixture test distributions used in simulated investigations of our estimator. Populations formed using $q_2 \in [0, 1)$ admixtures of $(1 - q_2)$exponential($\lambda = 5^{-1}$) and $q_2$Weibull($k = 5$, $\lambda = 1$) event time distributions. Event time density functions $\pi_t$ and corresponding survival functions $S_t$ are shown for various values of $q_2$.

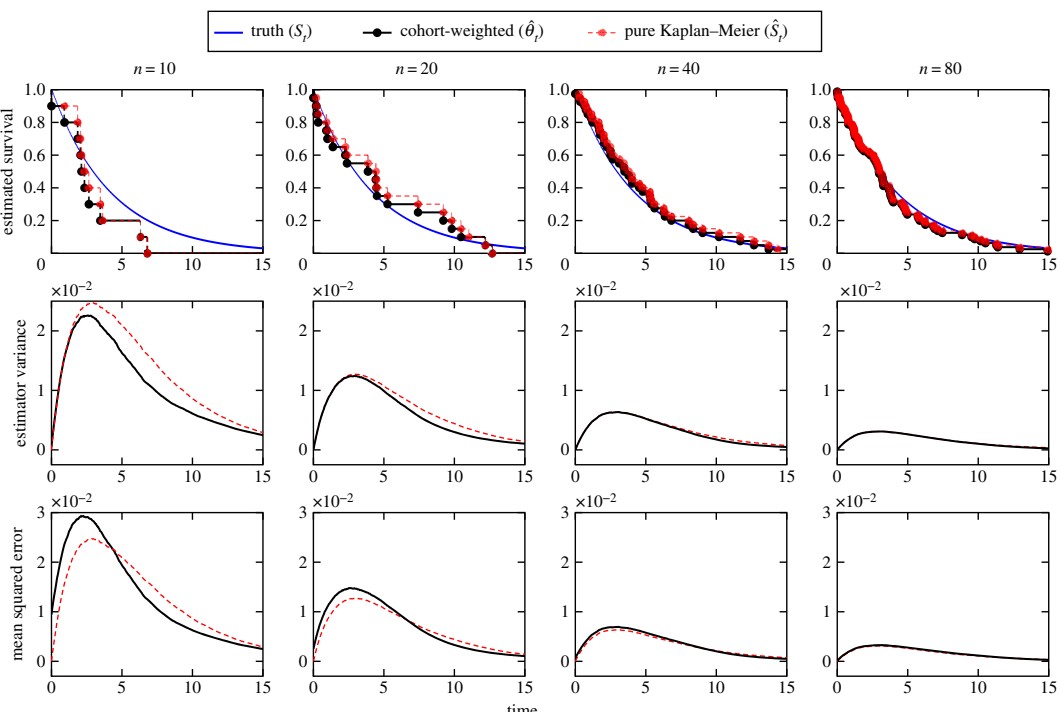

**Figure 3.** Comparing estimators of survival. The survival estimation method of equation (2.2) compared to pure Kaplan–Meier for a population containing an admixture of $(1 - q_2)$exponential($\frac{1}{5}$) and $q_2$Weibull(1, 5) individuals, where $q_2 = 0.25$. At a given sample size $n$, the survival estimates are obtained (top row: examples shown and contrasted). The estimator variance and mean square error were approximated using 10 000 resamplings for each of the sample sizes.

To compare the reweighted Kaplan–Meier estimator (equation (2.2)) to the standard Kaplan–Meier estimator, we estimated survival for the admixed population for $q_2 = 0.25$, using various sample sizes. In figure 3, we present example reconstructions using these two methods. The estimator variance was approximated using 10 000 resamplings of sample size $n$ of the admixed population, for each value

rsos.royalsocietypublishing.org    R. Soc. open sci. **5**: 180496

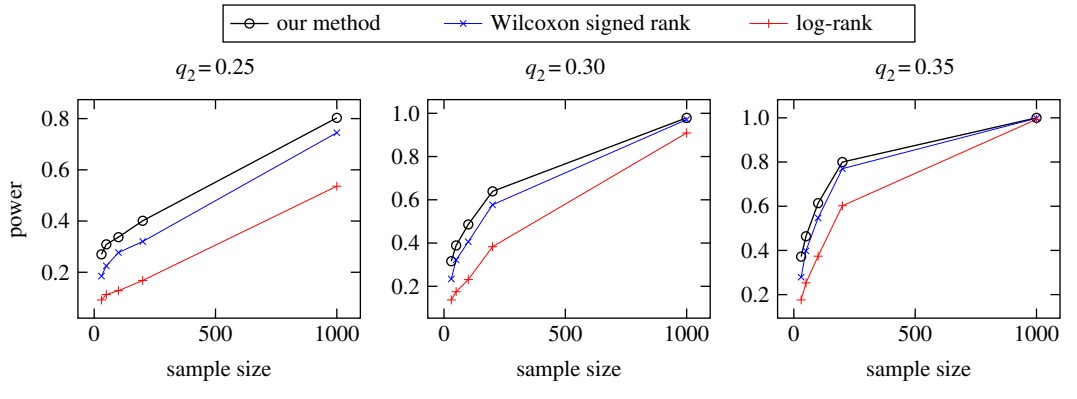

**Figure 4.** Simulated power computation comparing exponentially distributed lifetimes against a mixture of $q_2$ Weibull and $(1 - q_2)$ exponential distributions, where $q_2$ determines the amount of mixing. A larger value of $q_2$ implies more real difference between the survival functions of the two populations. The power of our method (black) is compared to the power of Kaplan–Meier Wilcoxon signed rank (blue) and log-rank (red) methods. (More power is better.)

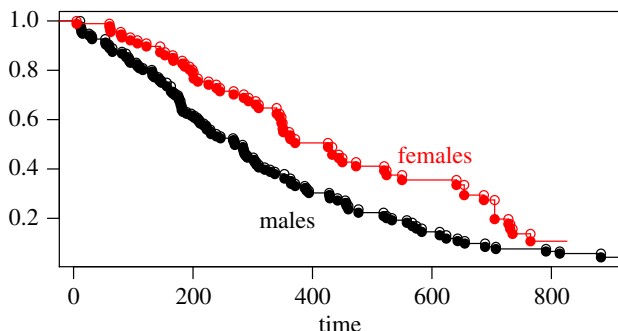

**Figure 5.** $\hat{\theta}_t$ estimates for days of lung cancer survival in males (population 1) versus females (population 2) from the NCCTG lung cancer dataset. The statistic $\hat{\Theta}$ implies an asymptotic $p$-value of 0.0009, rejecting $H_0$ at $\alpha = 0.05$.

of $n$. The estimation error, as defined by mean-squared difference between the reconstruction and the true survival function, was approximated in the same manner.

To better understand the performance of the test statistic (equation (2.4)), we evaluated its statistical power against that of other test statistics in distinguishing between Population 1 and Population 2 for various values of $q_2$. For samples of size $n^{(i)} \in \{30, 50, 100, 200, 1000\}$ taken from each population, we performed 1000 null hypothesis statistical tests using our method, the log-rank method [15], and the standard Kaplan–Meier Wilcoxon signed-rank difference-of-mean methods [16,17]. The power of the test, or the proportion of times that the null hypothesis was correctly rejected, is shown in figure 4.

## 3.2. Evaluating type I error in a real world example

We applied the survival estimator and statistic to NCCTG Lung Cancer data [18] available within the `survival` package for R. We compared the survival between male ($n^{(1)} = 136$) and female ($n^{(2)} = 90$) cancer patients, organized by ECOG performance score ($z \in \{0, 1, 2\}$) as cohort. Using males as population 1 and females as population 2, we arrived at the test-statistic estimate: $\hat{\Theta} = -961$, with 95% asymptotic confidence interval: $(-1527, -396)$, which would support rejection ($p \approx 0.0009$) of the null hypothesis ($\hat{\theta}_t^{(1)} = \hat{\theta}_t^{(2)}$) at $\alpha = 0.05$. For reference, both the Wilcoxon ($p \approx 0.0012$) and log-rank ($p \approx 0.0015$) tests referenced in figure 5 also rejected the null hypothesis. In figure 5, cohort-level survival estimates are also shown.

In theory, the type I error is set by the significance level at study design. Whether a statistic controls type I error correctly depends on accurate evaluation of its sampling distribution. In the case of $\hat{\Theta}$, our main result is that the sampling distribution for this estimator converges asymptotically in distribution to a Gaussian with a definite variance. However, small-sample behaviour is not guaranteed. To evaluate type I error, we used the same dataset, restricted to male patients. For each of $n \in \{40, 80, 136\}$, we sampled without replacement the $n$ male patients split into two groups so that $n^{(1)} = n^{(2)} = n/2$, and compared survival between the two random groups. Repeating this procedure 10 000 times,

rsos.royalsocietypublishing.org    R. Soc. open sci. 5: 180496

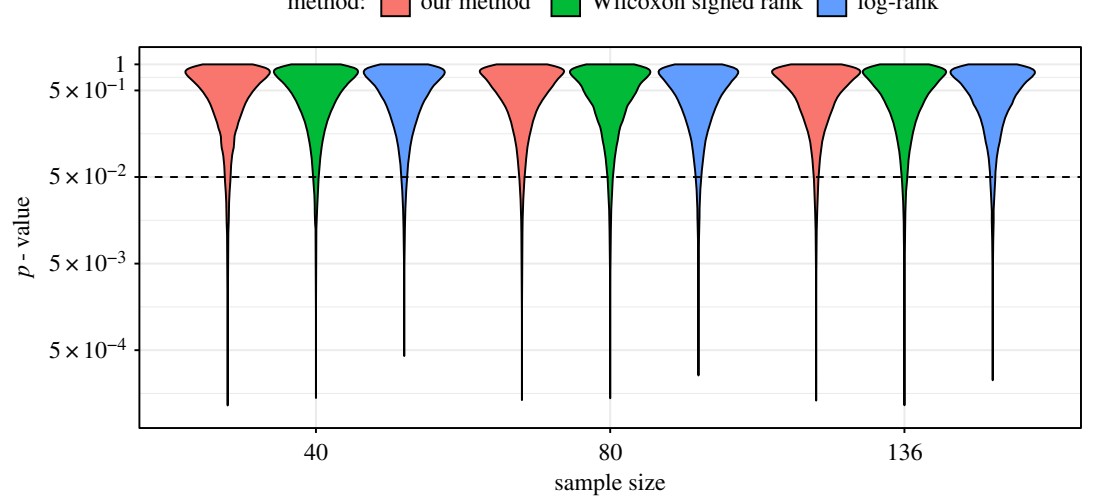

**Figure 6.** *P*-value distributions for the comparison between samples of size *n*/2 of two random subpopulations of male patients in the lung cancer data. The proportion of null hypotheses rejected for each of the three statistical methods is similar, at approximately 5% for $\alpha = 0.05$.

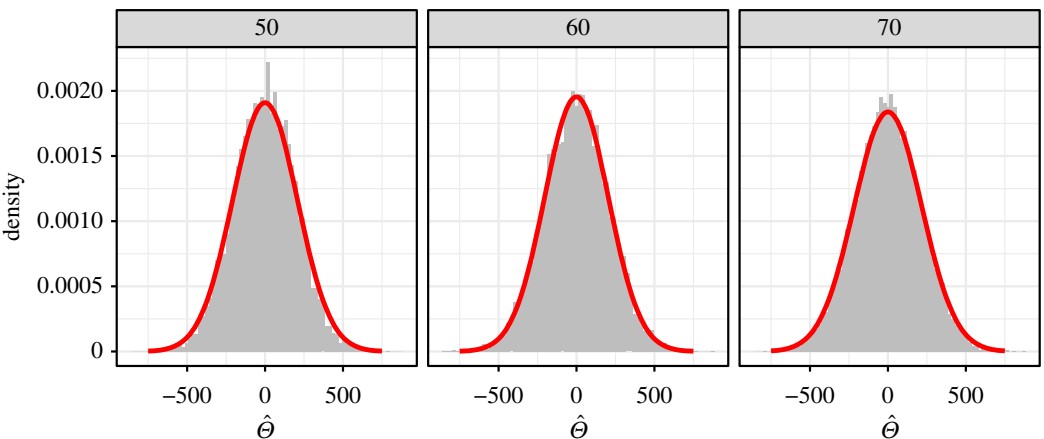

**Figure 7.** Histograms of $\hat{\Theta}$ sampling distributions for comparing survival between random subsets of male lung cancer patients using sample sizes of $n \in \{50, 60, 70\}$. Traced in red, the asymptotic Gaussian density as computed using theorem 2.1 on the first sample set of each size is overlayed.

we generated the observed distribution of *p*-values, presented in figure 6 in log-scale. The distributions computed using the three methods are similar. The three methods all rejected $H_0$ approximately 5% of the time except for the case of $\hat{\Theta}$ at $n = 40$, which rejected $H_0$ approximately 6% of the time. Essentially, asymptotic convergence as defined by the accurate evaluation of $\alpha = 0.05$ type I error occurs somewhere between 40 and 80 samples for this particular dataset.

Probing deeper, we examined the sampling distributions of $\hat{\Theta}$ for each of $n \in \{50, 60, 70\}$, in each instance compared to the Gaussian distribution stated in theorem 2.1, where the approximation is computed using only the first sample of size *n*. The results for these simulations are shown in figure 7, where it is seen that the sampling distribution of $\hat{\Theta}$ is approximately the same as the computed asymptotic Gaussian distribution, which is traced out in red.

The R code used to compute these examples is available in appendix B.3.

## 4. Discussion and conclusion

In this paper, we have proposed a test statistic that uses a cohort-averaged survival function estimator in order to make cross-population comparisons of survival within a null hypothesis statistical testing

framework. The proposed survival estimator was an empirically weighted average of cohort-level product-limit estimates. The test statistic involved computation of the area between estimated survival functions for two populations. By invoking an empirical stochastic process, we proved asymptotic normality of this test statistic.

Using simulations, we contrasted the weighted survival estimator against the pure Kaplan–Meier estimator. It is seen, in figure 3, that the survival curves generated from the two methods are distinct yet similar. In the second and third rows of figure 3, one sees that this reweighted estimator has comparable performance to the pure Kaplan–Meier estimator at large sample sizes. Asymptotically, both estimators converge to the true survival function, with variance converging to zero. At small sample sizes, there are differences. The reweighted estimator has reduced variance at the cost of larger bias, in a time-dependent manner. It also appears to have smaller variance at the cost of larger error at earlier times. This error at earlier times is mitigated by decreased error at later times (better reconstruction of tails); however, the estimator variance is lower at all times. Hence, dependent on costs, for small samples, this reweighted estimator may be preferable to the pure Kaplan–Meier estimator.

In simulations of the test statistic derived from the reweighted survival estimator, we saw superior performance compared to existing methods. In figure 4, it is seen that in all cases, the test statistic $\hat{\Theta}$ was better at distinguishing between the two populations than either the Wilcoxon signed-rank test or the log-rank test. The relatively high statistical power of this statistic is due to tighter variation in the test statistic. In nearly all cases (greater than 99.5%), the estimator variance for the tested method was less than that of the other two tests (not shown).

This paper derives the asymptotic convergence in distribution of the $\hat{\Theta}$ statistic. Numerically, we demonstrated convergence of the statistic in figures 6 and 7, where we verified that the asymptotic approximation respects type I error at $\alpha = 0.05$ and where we observe good match between the sampling distribution of $\hat{\Theta}$ and the asymptotic Gaussian distribution provided by theorem 2.1.

A variant of this method was used in Rasch et al. [9] in order to classify physical disorders based on severity for the sake of prioritization of processing for disability claims. Since the underlying survival surface is non-stationary, and the fixed observation windows create progressive censoring, that paper illustrates the utility of this statistical method. In that paper, the cohorts were defined based on binned application entry times and a heuristic 'survival surface' was generated in order to get a single overall picture of the survivability of a given disorder. The censoring parameters $\tau_z$ varied due to the finite sampling window and the fact that more recent cohorts are not observed for as long a time period as older cohorts, as depicted in figure 1b. It was also expected that survival by cohort would vary due to differences in healthcare administration and treatment between entry cohorts. The use of the empirical prevalences ($\hat{q}_z$) allowed the accounting for variability in disability application volume by sufferers of given disorders, conditional on entry date.

We note that a strong limitation of the presented method lies in its framing in terms of null hypothesis statistical testing. The $\hat{\Theta}$ statistic only provides a p-value, as opposed to other tests such as the log-rank test which provide hazard ratios as well. As a trade-off for statistical power, one is sacrificing interpretability in the form of effect sizes.

Although the most direct and natural applications of the method that we have presented here involve discretely indexed covariates, it is possible to use this method for continuously indexed covariates such as time by employing the binning strategy used by Rasch et al. [9]. This approach is particularly fruitful if the sampling windows are coarse, and there is a clear separation between cohorts to maintain statistical independence. In this situation, it may be unreasonable to expect to construct a full continuous surface for survival. Nonetheless, a possible future extension of this method might involve replacing the sum of equation (2.1) with an integral and using statistical regularization tools [19] in order to infer true continuously indexed surfaces.

Data accessibility. All data in this paper are simulated, with R source code provided in appendix B.2.

Authors' contributions. A.H. and M.H. developed the statistical method. A.H., M.H. and J.C.C. wrote the proof, performed the simulations and wrote the manuscript. J.C.C generated the figures. All authors gave final approval for publication.

Competing interests. The authors declare no competing interests.

Funding. This work is supported by the Intramural Research Program of the National Institutes of Health Clinical Center and the US Social Security Administration.

Acknowledgements. The authors thank Dr Leighton Chan and Dr Elizabeth Rasch for insightful discussions, guidance and support, Dr Pei-Shu Ho for help obtaining data.

# Appendix A. Proof of the main theorem

To prove the main theorem, we use an empirical process modelling framework to develop the asymptotic properties of first deterministically proportionally weighted Kaplan–Meier estimators. We then replace the deterministic proportions with estimates given by the sample prevalences of the cohorts. Here, we restate the main theorem and prove it through a series of lemmata.

**Theorem 2.1.** *Let $C_{z,t}^{(i)}$ denote the probability that a z-type individual has not yet been censored at time $t \geq 0$ (the survival probability relative to the occurrence of censoring), and $q_z^{(i)}$ denote the probability that an individual in population i is of cohort z, and let $p^{(i)} = n^{(i)}/(n^{(1)} + n^{(2)})$. Suppose that $\theta_t^{(1)} = \theta_t^{(2)}$. Then $\hat{\Theta} \xrightarrow{d} N(0, \sigma^2)$, as $n^{(i)} \to \infty$, with*

$$\sigma^2 = \sum_{i=1}^{2} (1 - p^{(i)}) \left( \sum_{z=1}^{d} q_z^{(i)} \phi_z^2 - \left( \sum_{z=1}^{d} q_z^{(i)} \phi_z \right)^2 \right) - \sum_{z=1}^{d} \int_0^{\tau_z} dS_{z,t} W_{z,t} \times \left( \frac{\phi_{z,t}}{S_{z,t}} \right)^2,$$

*where for $0 \leq t \wedge \tau_z$, where $\tau_z$ is the time at which samples of cohort z are censored, $\phi_{z,t} = \int_t^{\tau_z} dsS_{z,s}$, $\phi_z \equiv \phi_{z,0}$, $S_{z,t}$ is the survival function for the pooled data of cohort z, and*

$$W_{z,t} = \left( \frac{p^{(1)} C_{z,t-}^{(1)} q_z^{(2)} + p^{(2)} C_{z,t-}^{(2)} q_z^{(1)}}{C_{z,t-}^{(1)} C_{z,t-}^{(2)}} \right).$$

*The variance $\sigma^2$ may be consistently estimated by*

$$\hat{\sigma}^2 = \sum_{i=1}^{2} (1 - p^{(i)}) \left( \sum_{z=1}^{d} \hat{q}_z^{(i)} \hat{\phi}_z^2 - \left( \sum_{z=1}^{d} \hat{q}_z^{(i)} \hat{\phi}_z \right)^2 \right) - \sum_{z=1}^{d} \int_0^{\tau_z} d\hat{S}_{z,t} \hat{W}_{z,t} \times \left( \frac{\hat{\phi}_{z,t}}{\hat{S}_{z,t}} \right)^2, \tag{A 1}$$

*where for $0 \leq t \wedge \tau_z$, $\hat{S}_{z,t}$ is the product-limit estimate of the pooled data for cohort z*

$$\hat{\phi}_{z,t} = \int_t^{\tau_z} ds\hat{S}_{z,s}, \tag{A 2}$$

*$\hat{C}_{z,t}^{(i)}$ is the product-limit estimate associated with the event of censoring for cohort z within population i, $\hat{\phi}_z \equiv \hat{\phi}_{z,0}$, and*

$$\hat{W}_{z,t} = \left( \frac{p^{(1)} \hat{C}_{z,t-}^{(1)} \hat{q}_z^{(2)} + p^{(2)} \hat{C}_{z,t-}^{(2)} \hat{q}_z^{(1)}}{\hat{C}_{z,t-}^{(1)} \hat{C}_{z,t-}^{(2)}} \right). \tag{A 3}$$

*Overview of proof of theorem 2.1.* To prove the main theorem, we turn to the modelling framework that we present in A.2. In general, we proceed by first assuming fixed sample proportions and then extending results to random proportions as given by empirical prevalence (equation (A 2)). The convergence of $\hat{\Theta}$ follows directly from corollary A.9 and equation (A 18). The consistency of $\hat{\sigma}^2$ follows from theorem 4.2.2 of [2], which provides for weak convergence of the product limit estimator to a Gaussian process, and the Glivenko–Cantelli theorem. ∎

## A.1. Preliminaries and notation

Given any pair of random elements $X$, $Y$, we denote equality in a distributional sense by $X \approx Y$. Let $\mathbb{P}$ be a probability measure on the measurable space $(X, \mathcal{A})$. The empirical measure generated by the sample of random elements $x_1, \ldots, x_n$, $n \in \mathbb{N}$ is given by

$$\mathbb{P}_n = n^{-1} \sum_{i=1}^{n} \delta_{x_i}, \tag{A 4}$$

where for any $x \in X$, and any $A \in \mathcal{A}$,

$$\delta_x(A) = \begin{cases} 1, & x \in A, \\ 0, & x \notin A. \end{cases} \tag{A 5}$$

Note that alternatively, when needed, one may write $\delta_x(A)$ as the indicator function $1_A(x)$ on the set $A$. Furthermore, in the case that $A = \{k\}$, $k \in \mathbb{Z}$, and $x \in \mathbb{Z}$, we write $\delta_x(A) \equiv \delta_{x,k}$.

Given $\mathcal{H}$, a class of measurable functions $h: X \to \mathbb{R}$, the empirical measure generates the map $\mathcal{H} \to \mathbb{R}$ given by $h \mapsto \mathbb{P}_n h$, where for any signed measure $Q$ and measurable function $h$, we use the notation

$Qh = \int \mathrm{d}Qh$. Furthermore, define the $\mathcal{H}$-indexed empirical process $\mathcal{G}_n$ by

$$\mathcal{G}_n h = \sqrt{n}(\mathbb{P}_n - \mathbb{P})h = \frac{1}{\sqrt{n}} \sum_{i=1}^{n} (h(x_i) - \mathbb{P}h), \tag{A 6}$$

and with the empirical process, identify the signed measure $\mathcal{G}_n = n^{-1/2} \sum_{i=1}^{n} (\delta_{x_i} - \mathbb{P})$.

Note that for a measurable function $h$, from the law of large numbers and the central limit theorem, it follows that $\mathbb{P}_n h \xrightarrow{a.s.} \mathbb{P}h$, and $\mathcal{G}_n h \xrightarrow{d} N(0, \mathbb{P}h - \mathbb{P}h^2)$, provided $\mathbb{P}h$ exists and $\mathbb{P}h^2 < \infty$, and where '$\xrightarrow{d}$' denotes convergence in distribution. In addition to the preceding notation, given the elements $f$, and $f_n$, $n \in \mathbb{N}$, we also denote, respectively, convergence in probability and in distribution, of $f_n$ to $f$, by $f_n \xrightarrow{p} f$.

For any map $x : \mathcal{H} \to \mathbb{R}^k$, $k \in \mathbb{N}$, define the uniform norm $\|x\|_{\mathcal{H}}$ by

$$\|x\|_{\mathcal{H}} = \sup \{|x(h)| : h \in \mathcal{H}\}, \tag{A 7}$$

and in the case that $\mathcal{H} \subset \mathbb{R}$, write $\|\cdot\|_{\mathcal{H}} \equiv \|\cdot\|_{\infty}$. A class $\mathcal{H}$ for which $\|\mathbb{P}_n - \mathbb{P}\|_{\mathcal{H}} \to 0$ is called a $\mathbb{P}$-Glivenko–Cantelli class. Denote by $\ell^{\infty}(\mathcal{H})$ the class of uniformly bounded functions on $\mathcal{H}$. That is, for a general $k \in \mathbb{N}$,

$$\ell^{\infty}(\mathcal{H}) = \{x : \mathcal{H} \to \mathbb{R}^k : \|x\|_{\mathcal{H}} < \infty\}.$$

If for some tight Borel measurable element $\mathcal{G} \in \ell^{\infty}(\mathcal{H})$, $\mathcal{G}_n \xrightarrow{d} \mathcal{G}$, in $\ell^{\infty}(\mathcal{H})$, we say that $\mathcal{H}$ is a $\mathbb{P}$-Donsker class.

## A.2. Empirical process framework

To prove theorem 2.1, we turn to an empirical modelling framework that will provide us the asymptotic statistics of the weighted product limit estimator. Consider a closed particle system, such that according to a predefined set of characteristics, the system can be subdivided into mutually exclusive subsystems.

Each particle corresponds to the observed state of a particular individual in a fixed population cohort. Note that we will restrict this discussion to only a single population of particles. These arguments will extend to multiple populations as mentioned in this paper by treating separate populations as independent.

At any given time $t \geq 0$, each particle will have exactly one associated state $x$ in the set $\mathbb{Z}_4$, referring, respectively, to states of

$$\left. \begin{array}{ll} 0 & \text{dormancy} \\ 1 & \text{activity} \\ 2 & \text{inactivity} \\ 3 & \text{censored.} \end{array} \right\} \tag{A 8}$$

Assume that the path of any particle is statistically dependent upon its particular subsystem, and that given the respective subsystems of any two particles, their resulting paths are statistically independent. Assume further that at a reference time $t = 0$, all particles enter into the active state ($x = 1$), and that particles are considered dormant for all $t < 0$.

Let $d \in \mathbb{N}$ and $\tau \in (0, \infty)$ be fixed. We will assume the existence of a collection of individuals $\Gamma$, assumed to be infinite in size, where each individual $\gamma \in \Gamma$ exhibits a càdlàg path-valued state $x_t^{\gamma}$, for $t \geq 0$. For each $\gamma \in \Gamma$, $x_t^{\gamma}$ is determined by the individual's particle type $z^{\gamma}$ and a random jump time $\xi^{\gamma}$. The particle type $z^{\gamma}$ is distributed in the population through the probability mass $\mathbb{P}(z^{\gamma} = z) = q_z$, where $\mathbf{q} = (q_1, , q_d) \in (0, 1)^d$ satisfies $\sum_{z=1}^{d} q_z = 1$. Let $S_t = (S_{1,t}, \ldots, S_{d,t})$ be the survival vector $S_{z,t} = \mathbb{P}\{T_z > t\}$, which is assumed continuous for $t \geq 0$. Suppose that it is desired to understand the event probabilities for randomly selected $\gamma \in \Gamma$, unconditional on subgroup membership. We assume that members of each cohort are in the inactive (0) state at times $t < 0$.

Given a random sample $\gamma_1, \ldots, \gamma_n$, $n \in \mathbb{N}$ of individuals, let $\mathbf{n} = (n_1, \ldots, n_d)$ and

$$n = \sum_{z=1}^{d} n_z, \tag{A 9}$$

where $n_z$ is the random number of drawn individuals of cohort $z$. In considering the event time probabilities of each subgroup, the random number of particles excludes the use of many well-established results in survival analysis. Therefore, we begin with a somewhat restricted framework, and assume a known number of initial individuals of each type.

Assume the sample contains a *known* number $n_z = a_z n$, $a_z \in (0, 1)$, of individuals of cohort $z$, and let $\mu_{j,z,t}^{n_z} \geq 0$ be the number of the cohort $z$ individuals who are in state $j \in \mathbb{Z}_4$ at time $t$, so that

$$\sum_{j=0}^{3} \mu_{j,z,t}^{n_z} = n_z \tag{A 10}$$

is conserved. Also, we assume that there exists $\tau_z < \infty$ when all particles either become inactive or censored so that $\tau_z$ is the infimum time where the condition

$$n_z = \mu_{2,z,t}^{n_z} + \mu_{3,z,t}^{n_z} \quad \forall t > \tau_z \tag{A 11}$$

holds.

For the sample of size $n_z$, we denote the $z$-type cumulative hazard by $\Lambda_{z,t}$ and, respectively, define the $z$-type cumulative hazard and survival estimates by

$$\hat{\Lambda}_{z,t}^{n_z} = \int_0^t \frac{d\mu_{2,z,s}^{n_z}}{\mu_{1,z,s-}^{n_z}} \tag{A 12}$$

and

$$\hat{S}_{z,t}^{n_z} = \prod_{s \le t} (1 - d\hat{\Lambda}_{z,s}^{n_z}). \tag{A 13}$$

Define further

$$B_{z,t}^{n_z} = \sqrt{n_z} \frac{\hat{S}_{z,t}^{n_z} - S_{z,t}}{S_{z,t}}$$

and note that $\hat{S}_{z,t}^{n_z} = \hat{S}_{z,\tau_z}^{n_z}$ and $B_{z,t}^{n_z} = B_{z,\tau_z}^{n_z}$ for all $t \ge \tau_z$.

From [2], it follows that $\{B_{z,t}^{n_z} : t \ge 0\}$ is a mean-zero square-integrable martingale with Meyer bracket process

$$\langle B_{z,t}^{n_z}, B_{w,t}^{n_z} \rangle_t = \delta_{zw} n_z \int_0^{t \wedge \tau_z} d\Lambda_{z,s} \left( \frac{\hat{S}_{z,s-}^{n_z}}{S_{z,s}} \right)^2 \frac{1_{\{\mu_{1,z,s-}^{n_z} > 0\}}}{\mu_{1,z,s-}^{n_z}}, \tag{A 14}$$

where $t \wedge \tau_z = \min\{t, \tau_z\}$, and $\delta_{(\cdot,\cdot)}$ is the Kroenicker delta function.

## A.3. Convergence theorems

In order to guarantee convergence of the estimator, we make the following assumptions (based upon an initially known sample size distribution **n**).

**Assumption A.1.** *We assume that the initial sample is chosen large enough to ensure that individuals of cohort z, at state 1 (active), exist at all points $t \in [0, \tau_z]$, $z \in \{1, \ldots, d\}$. That is,*

$$\inf_{z \in \mathbb{N}_d} \mu_{1,z,\tau_z-}^{n_z} > 0, \quad \text{a.s.}$$

Since any survival function is monotone, an immediate result that follows from the above assumption is

$$c < S_{z,\tau_z} \le S_{z,t} \le 1, \quad t \ge 0, \tag{A 15}$$

for some constant $c > 0$.

**Assumption A.2.** *It is assumed that as n becomes large, the sample size for each individual type will grow to infinity. That is,*

$$\lim_{n \to \infty} \inf_{z \in \mathbb{N}_d, a \in V} \mu_{1,z,\tau_z-}^{na_z} = \infty, \quad \text{a.s.}$$

**Assumption A.3.** *For each $z \in \{1, \ldots, d\}$ there exists a non-increasing continuous function $m_z : [0, \infty) \to (0, 1]$ such that*

$$\lim_{n \to \infty} \sup_{t \ge 0} \left| \frac{\mu_{1,z,t}^{na_z}}{na_z} - m_{z,t} \right| = 0 \quad \text{a.s.}$$

Note that in the case of fixed censoring, that is, in the case that censoring exists only at time $\tau$, the above is satisfied by $m_{z,t} = S_{z,t}$. In the general case, $m_{z,t}$ can be seen as the probability that an individual of cohort $z$ has not yet left state 1. That is, $m_{z,t}$ is the probability that an individual has not left due to censoring or death by time $t$, and so $m_{z,t} = S_{z,t} C_{z,t-}$, where $C_{z,t}$ is the probability that censoring has not occurred by time $t$.

To prove the main theorem, we now present a series of lemmata.

**Lemma A.4.** *If $\hat{q}$ is defined as in equation (2.3) and $\hat{S}_{z,s-}^{n_z}$ is defined as in equation (A 13), then*

$$\sqrt{n} \sum_{z=1}^{d} (\hat{q}_z - q_z) \int_0^{t \wedge \tau_z} ds (\hat{S}_{z,s-}^{n\hat{q}_z} - S_{z,s}) \xrightarrow{p} 0,$$

*as $n \to \infty$, uniformly in $t \geq 0$.*

*Proof.* It is claimed that to prove the statement of the lemma, it suffices to show that

$$\sup_{t \geq 0} \left( \frac{\hat{S}_{z,t-}^{n\hat{q}_z} - S_{z,t}}{S_{z,t}} \right)^2 \xrightarrow{p} 0, \tag{A 16}$$

uniformly in $t \geq 0$, for each $z = 1, \ldots, d$.

Indeed, for if the above holds, then

$$\int_0^{t \wedge \tau_z} ds (\hat{S}_{z,s-}^{n\hat{q}_z} - S_{z,s}) \xrightarrow{p} 0,$$

uniformly in $t \geq 0$. Since the central limit theorem implies that $\sqrt{n}(\hat{q}_z - q_z) \xrightarrow{d} N(0, q_z(1 - q_z))$, each term in the sum would converge in probability to 0, uniformly in $t \geq 0$.

And so, if $\mathbb{E}_N$ denotes the expectation given $N$, we have that

$$\mathbb{E} \left( \frac{\hat{S}_{z,t-}^{n\hat{q}_z} - S_{z,t}}{S_{z,t}} \right)^2 = \mathbb{E} \frac{1}{n\hat{q}_z} \mathbb{E}_{n\hat{q}_z} (B_{z,t}^{n\hat{q}_z})^2 = \mathbb{E} \frac{1}{n\hat{q}_z} \mathbb{E}_{n\hat{q}_z} n\hat{q}_z \int_0^{t \wedge \tau_z} \frac{d\Lambda_{z,s}}{\mu_{1,z,s-}^{n\hat{q}_z}} \left( \frac{\hat{S}_{z,s-}^{n\hat{q}_z}}{S_{z,s}} \right)$$

$$= \mathbb{E} \int_0^{t \wedge \tau_z} \frac{d\Lambda_{z,s}}{\mu_{1,z,s-}^{n\hat{q}_z}} \left( \frac{\hat{S}_{z,s-}^{n\hat{q}_z}}{S_{z,s}} \right) \leq C \mathbb{E} (\mu_{1,z,\tau_z}^{n\hat{q}_z})^{-1},$$

for some arbitrary constant $C$. From Lenglart's inequality (cf. [20]),

$$\mathbb{P} \left\{ \sup_t \left( \frac{\hat{S}_{z,t-}^{n\hat{q}_z} - S_{z,t}}{S_{z,t}} \right)^2 > \epsilon \right\} \leq \frac{\eta}{\epsilon} + \mathbb{P} \left\{ \mu_{1,z,\tau_z-}^{n\hat{q}_z} < \frac{C}{\eta} \right\},$$

for any arbitrary $\eta, \epsilon > 0$. Therefore, from assumption A.2, since $n_z \to \infty$ a.s., the desired result follows. ∎

Turning momentarily to the situation where there are two populations denoted by superscripts (1) and (2), for any $t \geq 0$, define

$$\hat{\Theta}_t^{\delta} = \sqrt{\frac{n^{(2)}}{n^{(1)} + n^{(2)}}} \int_0^{t \wedge \tau} ds \sqrt{n^{(1)}} (\hat{\theta}_{s-}^{(1)} - \theta_s^{(1)}) - \sqrt{\frac{n^{(1)}}{n^{(1)} + n^{(2)}}} \int_0^{t \wedge \tau} ds \sqrt{n^{(2)}} (\hat{\theta}_{s-}^{(2)} - \theta_s^{(2)}),$$

noting that setting $\theta_s^{(1)} = \theta_s^{(2)}$ recovers our test statistic of equation (2.4). For a general survival function $\theta$, with respective estimate $\hat{\theta}$, define $\hat{Y}_t$ by

$$\hat{Y}_t = \int_0^{t \wedge \tau} ds \sqrt{n} (\hat{\theta}_{s-} - \theta_s), \quad t \geq 0. \tag{A 17}$$

If the process $\hat{Y}$ converges in distribution to some $Y \sim N(0, \sigma^2)$, since $n^{(i)}/(n^{(1)} + n^{(2)})$ converges to $p^{(i)}$, $i = 1, 2$, it follows that

$$\hat{\Theta}_t^{\delta} \xrightarrow{d} \sqrt{p^{(2)}} Y_t^{(1)} - \sqrt{p^{(1)}} Y_t^{(2)} \approx N(0, p^{(2)} \sigma_1^2 + p^{(1)} \sigma_2^2). \tag{A 18}$$

Now we turn to analysis under a single population, dropping the superscripts. Note that $\hat{Y}_t = \sum_{z=1}^{d} \hat{Z}_{z,t}$, where

$$\hat{Z}_{z,t} = \sqrt{n} \int_0^{t \wedge \tau_z} ds (\hat{q}_z \hat{S}_{z,s-}^{n\hat{q}_z} - q_z S_{z,s}) = \sqrt{n} (\hat{q}_z - q_z) \int_0^{t \wedge \tau_z} ds (\hat{S}_{z,s-}^{n\hat{q}_z} - S_{z,s}) + \sqrt{n} (\hat{q}_z - q_z) \int_0^{t \wedge \tau_z} ds S_{z,s}$$

$$+ \sqrt{n} q_z \int_0^{t \wedge \tau_z} ds (\hat{S}_{z,s-}^{n\hat{q}_z} - S_{z,s}). \tag{A 19}$$

Therefore, if it can be shown that

$$\sqrt{n}\sum_{z=1}^{d}(\hat{q}_z - q_z)\int_0^{t\wedge\tau_z} ds(\hat{S}_{z,s-}^{n\hat{q}_z} - S_{z,s}) \xrightarrow{p} 0,$$

uniformly in $t$, then convergence of $(\hat{Y}_t : t \geq 0)$ is dependent only upon the convergence of the $d$-dimensional vector-valued process $\hat{\zeta}(\hat{q})$ given by

$$\hat{\zeta}_{z,t}(a) = \sqrt{n}(\hat{q}_z - q_z)\int_0^{t\wedge\tau_z} ds\, S_{z,s} + \sqrt{n}q_z\int_0^{t\wedge\tau_z} ds\,(\hat{S}_{z,s-}^{na_z} - S_{z,s}), \tag{A 20}$$

with $a = (a_1, \ldots, a_d) \in (0,1)^d$ chosen in a sufficiently small neighbourhood $V$ of $q$. This decomposition will thus lead to the main theorem. To show the desired convergence of $\hat{\zeta}_t(\hat{q})$, we first focus on convergence of $\hat{\zeta}_t(a)$.

Let $\phi_{z,t} = \int_t^{\tau_z} ds S_{z,s}$ and write $\hat{\zeta}_t(a) = \hat{\zeta}_t^1 + \hat{\zeta}_t^2(a)$, where

$$\hat{\zeta}_{z,t}^1 = \sqrt{n}(\hat{q}_z - q_z)\int_0^{t\wedge\tau_z}(-d\phi_{z,s}) \tag{A 21}$$

and

$$\hat{\zeta}_{z,t}^2(a) = \frac{q_z}{\sqrt{a_z}}\int_0^{t\wedge\tau_z}(-d\phi_{z,s})B_{z,s}^{na_z}. \tag{A 22}$$

**Lemma A.5.** *Suppose that $\{\hat{\zeta}_t^1(a) : t \geq 0\}$ and $\{\hat{\zeta}_t^2(a) : t \geq 0\}$ are the processes respectively defined by equations (A 21) and (A 22), and that $\tilde{B}$ is the $d$-dimensional mean-zero Gaussian process defined by*

$$\langle \tilde{B}_z, \tilde{B}_w \rangle_t = \delta_{z,w}\int_0^{t\wedge\tau_z}\frac{d\Lambda_{z,s}}{S_{z,s}C_{z,s-}}. \tag{A 23}$$

*Then $\hat{\zeta}_t^1 \xrightarrow{d} \zeta_t^1$ and $\hat{\zeta}_t^2(a) \xrightarrow{d} \zeta_t^2(a)$, in the space of compactly supported functions $\mathcal{D}_{\mathbb{R}^d}[0,\infty)$ as $n \to \infty$, for each $a \in V$, where $\zeta_t^1 = (\zeta_{1,t}^1, \ldots, \zeta_{d,t}^1)$ is the mean-zero square-integrable Gaussian process defined by*

$$\langle \zeta_z^1, \zeta_w^1 \rangle_t = -q_z q_w\left(\int_0^{t\wedge\tau_z} ds S_{z,s}\right)\left(\int_0^{t\wedge\tau_w} ds S_{w,s}\right) + \delta_{z,w}q_z\left(\int_0^{t\wedge\tau_z} ds\, S_{z,s}\right)^2, \tag{A 24}$$

*and $\zeta_t^2(a) = (\zeta_{1,t}^2(a), \ldots, \zeta_{d,t}^2(a))$ is given by*

$$\zeta_{z,t}^2(a) = \frac{q_z}{\sqrt{a_z}}\left(\int_0^{t\wedge\tau_z} d\tilde{B}_{z,s}\phi_{z,s} - \phi_{z,t\wedge\tau_z}\tilde{B}_{z,t\wedge\tau_z}\right). \tag{A 25}$$

*The processes $\hat{\zeta}^1$ and $\hat{\zeta}^2(a)$ are independent, and there exist Skorohod representations such that*

$$\sup_{t\geq 0}|\hat{\zeta}_{z,t}^1 - \zeta_{z,t}^1| \to 0$$

*and*

$$\sup_{t\geq 0, a\in V}|\hat{\zeta}_{z,t}^2(a) - \zeta_{z,t}^2(a)| \to 0,$$

*almost surely as $n \to \infty$.*

*Proof.* To begin note that independence follows immediately from the independence of the respective limiting processes. Since $\mathbf{n}$ is a multinomial random variable, (A 24) follows from the central limit theorem. In the case of $\hat{\zeta}_t^2(a)$, we first consider $B_{z,t}^{na_z}$.

An application of Lenglart's inequality, very similar to that in the proof of lemma A.4, along with assumption A.2, shows that

$$\sup_{a\in V, t\geq 0}|\hat{S}_{z,t-}^{na_z} - S_{z,t}| \xrightarrow{p} 0, \quad \text{as } n \to \infty.$$

Moreover, from assumption A.3,

$$\sup_{a\in V, t\geq 0}\left|\frac{na_z}{\mu_{1,z,t-}^{na_z}} - \frac{1}{m_{z,t}}\right| \xrightarrow{p} 0, \quad \text{as } n \to \infty.$$

It follows that

$$\frac{na_z}{\mu_{1,z,t-}^{na_z}}\left(\frac{\hat{S}_{z,t-}^{na_z}}{S_{z,t}}\right)^2 \xrightarrow{p} \frac{1}{m_{z,t}},$$

uniformly in $t \geq 0$, and since $m_{z,t} = S_{z,t}\, C_{z,t-}$,

$$\langle B_{z,t}^{na_z}, B_{w,t}^{na_z}\rangle_t \xrightarrow{p} \delta_{z,w}\int_0^{t\wedge\tau_z}\frac{\mathrm{d}\Lambda_{z,s}}{S_{z,s}C_{z,s-}}.$$

Therefore, from theorem 4.2.1 of [2], $B_{z,t}^{na_z} \xrightarrow{d} \tilde{B}_{z,t}$, and there exists a Skorohod representation of $B_{z,t}^{na_z}$ such that

$$\sup_{t\geq 0, a\in V}|B_{z,t}^{na_z} - \tilde{B}_{z,t}| \to 0,$$

almost surely as $n \to \infty$. Since almost sure convergence of $B_{z,t}^{na_z}$ implies almost sure convergence of bounded functionals of $B_{z,t}^{na_z}$, the desired convergence of $\hat{\zeta}^2(a)$ follows from theorem 2.1 of [3]. ∎

**Corollary A.6.** *If the process $\hat{\zeta}(a) = \{\hat{\zeta}_t(a)\}$ is defined by equation (A 20), then*

$$\sum_{z=1}^{d}\hat{\zeta}_z(a) \xrightarrow{d} \sum_{z=1}^{d}\zeta_{z,t}(a) = \sum_{z=1}^{d}\zeta_{z,t}^1 + \zeta_{z,t}^2(a). \tag{A 26}$$

*Proof.* From the previous theorem, we may assume that $\hat{\zeta}_{z,t}^1 \to \zeta_{z,t}^1$ and $\hat{\zeta}_{z,t}^2(a) \to \zeta_{z,t}^2(a)$ almost surely, uniformly for $a \in V$ and $t \geq 0$. Therefore,

$$\hat{\zeta}_t(a) \to \zeta_t(a)$$

almost surely, uniformly for $a \in V$ and $t \geq 0$. The statement of the theorem then follows from theorem 5.1 of [21]. ∎

Since $\mathbf{n}/n \xrightarrow{p} \mathbf{q}$, from theorem 4.4 of [21]

$$\left(\frac{\mathbf{n}}{n}, \left\{\sum_{z=1}^{d}\hat{\zeta}_{z,t}(a) : t \geq 0\right\}\right) \xrightarrow{d} \left(\mathbf{q}, \left\{\sum_{z=1}^{d}\zeta_{z,t}(a) : t \geq 0\right\}\right).$$

Define the map $g : V \times \ell^\infty(V \times [0, \infty)) \to \ell^\infty([0, \infty))$ by $g(a, f) = f(a, \cdot\,)$, then

$$\sum_{z=1}^{d}\zeta_{z,t}\left(\frac{\mathbf{n}}{n}\right) = g\left(\frac{\mathbf{n}}{n}, \sum_{z=1}^{d}\zeta_z\right).$$

Furthermore, if for any $(a_1, f_1), (a_2, f_2) \in V \times \ell^\infty(V \times [0, \infty))$ we have that

$$|a_1 - a_2| + \sup_{a\in V, t\geq 0}|f_1(a, t) - f_2(a, t)| < \delta$$

for some $\delta > 0$, then

$$\sup_{t\geq 0}|g(a_1, f_1)(t) - g(a_2, f_2)(t)| = \sup_{t\geq 0}|f_1(a_1, t) - f_2(a_2, t)| \leq \sup_{t\geq 0}|f_1(a_1, t) - f_1(a_2, t)| + \sup_{t\geq 0}|f_1(a_2, t) - f_2(a_2, t)|.$$

Therefore, $g$ is continuous at any $(a, f)$ such that $f$ is continuous at $a$, uniformly in $t$. It thus follows from the continuous mapping theorem (cf. [4]) that if $a \mapsto \sum_{z=1}^{d}\zeta_{z,t}(a)$ is continuous, uniformly in $t$, then

$$g\left(\frac{\mathbf{n}}{n}, \sum_{z=1}^{d}\hat{\zeta}_z\right) \xrightarrow{d} g\left(\mathbf{q}, \sum_{z=1}^{d}\zeta_z\right). \tag{A 27}$$

**Lemma A.7.** *If $\{\zeta_t(a) : t \geq 0\}$ is defined as in corollary A.6, then the map*

$$a \mapsto \sum_{z=1}^{d}\zeta_{z,t}(a)$$

*is continuous for $a \in V$, uniformly in $t \geq 0$.*

*Proof.* For any $a, b \in V$, it follows that

$$\sum_{z=1}^{d} \zeta_{z,t}(a) - \sum_{z=1}^{d} \zeta_{z,t}(b) = \sum_{z=1}^{d} q_z \left( \frac{1}{\sqrt{a_z}} - \frac{1}{\sqrt{b_z}} \right) \times \left( \phi_{z,t \wedge \tau_z} \tilde{B}_{z,t \wedge \tau_z} - \int_0^{t \wedge \tau_z} \mathrm{d}\tilde{B}_{z,s} \phi_{z,s} \right).$$

Since $S_{\tau_z} > 0$ for all $z$, from Doob's martingale inequality (cf. [22]),

$$\mathbb{E} \sup_{t \geq 0} \left( \sum_{z=1}^{d} \zeta_{z,t}(a) - \sum_{z=1}^{d} \zeta_{z,t}(b) \right)^2 \leq C \sum_{z=1}^{d} \left( \frac{1}{\sqrt{a_z}} - \frac{1}{\sqrt{b_z}} \right)^2,$$

for some arbitrary constant $C$. For each $z \in \mathbb{N}_d$, since $a_z$ and $b_z$ are sufficiently close to $q_z \in (0, 1)$, it follows that there exists some $\delta > 0$ such that $a_z \wedge b_z > \delta$. Therefore,

$$\left( \frac{1}{\sqrt{a_z}} - \frac{1}{\sqrt{b_z}} \right)^2 = \frac{1}{a_z b_z} (\sqrt{a_z} - \sqrt{b_z})^2 \leq \delta^{-2} (\sqrt{a_z} - \sqrt{b_z})^2 \left( \frac{\sqrt{a_z} + \sqrt{b_z}}{\sqrt{a_z} + \sqrt{b_z}} \right)^2 \leq \frac{1}{4\delta^3} (a_z - b_z)^2.$$

Combining the above two results gives

$$\mathbb{E} \sup_{t \geq 0} \left( \sum_{z=1}^{d} \zeta_{z,t}(a) - \sum_{z=1}^{d} \zeta_{z,t}(b) \right)^2 \leq C |a - b|^2,$$

and so, by Kolmogorov's continuity criterion (cf. [22]), the desired result follows. ∎

The above lemma, along with the argument immediately preceding, gives the following.

**Theorem A.8.** *Let $\sum_{z=1}^{d} \zeta_{z,t}^n(\cdot)$ and $\sum_{z=1}^{d} \zeta_{z,t}(\cdot)$ be defined as in corollary A.6, then*

$$\sum_{z=1}^{d} \hat{\zeta}_{z,t} \left( \frac{\mathbf{n}}{n} \right) \overset{d}{\longrightarrow} \sum_{z=1}^{d} \zeta_{z,t}(\mathbf{q}), \quad \text{in } \mathcal{D}_{\mathbb{R}}[0, \infty), \quad \text{as } n \to \infty. \tag{A 28}$$

**Corollary A.9.** *If $\hat{\zeta} = \sum_{z=1}^{d} \zeta_{z,\tau_z}(q)$, then*

$$\hat{\zeta} \sim N(0, \sigma^2),$$

*where*

$$\sigma^2 = \sum_{z=1}^{d} q_z \phi_{z,0}^2 - \left( \sum_{z=1}^{d} q_z \phi_{z,0} \right)^2 - \sum_{z=1}^{d} q_z \int_0^{\tau_z} \frac{\mathrm{d}S_{z,t}}{C_{z,t-}} \left( \frac{\phi_{z,t}}{S_{z,t}} \right)^2.$$

*Proof.* Note that when $t = \tau_z$, we have

$$\zeta_{z,\tau_z}(q) = \zeta_{z,\tau_z}^1 + \sqrt{q_z} \int_0^{\tau_z} \mathrm{d}\tilde{B}_{z,t} \phi_{z,t},$$

which are independent and normally distributed, implying that $\hat{\zeta}$ is also normally distributed. Furthermore,

$$\mathbb{E}\hat{\zeta}^2 = \sum_{z=1}^{d} \left( \zeta_{z,\tau_z}^1 + \sqrt{q_z} \int_0^{\tau_z} \mathrm{d}\tilde{B}_{z,t} \phi_{z,t} \right)^2 + \sum_{\substack{z,w=1 \\ z \neq w}}^{d} \left( \zeta_{z,\tau_z}^1 + \sqrt{q_z} \int_0^{\tau_z} \mathrm{d}\tilde{B}_{z,t} \phi_{z,t} \right) \times \left( \zeta_{w,\tau_w}^1 + \sqrt{q_z} \int_0^{\tau_w} \mathrm{d}\tilde{B}_{w,t} \phi_{w,t} \right)$$

$$= \sum_{z=1}^{d} \left( \mathbb{E}\zeta_{z,\tau_z}^{1 \, 2} \right) + \mathbb{E}q_z \left( \int_0^{\tau_z} \mathrm{d}\tilde{B}_{z,t} \phi_{z,t} \right)^2 - \sum_{\substack{z,w=1 \\ z \neq w}}^{d} \mathbb{E}\zeta_{z,\tau_z}^1 \zeta_{w,\tau_w}^1 = \sum_{z=1}^{d} \left( q_z(1 - q_z) \phi_{z,0}^2 + q_z \int_0^{\tau_z} \mathrm{d}\Lambda_{z,t} \frac{\phi_{z,t}^2}{S_{z,t} C_{z,t-}} \right)$$

$$- \sum_{\substack{z,w=1 \\ z \neq w}}^{d} q_z q_w \phi_{z,0} \phi_{w,0},$$

which after recombining the final terms, gives the desired result. ∎

# Appendix B. Computation

## B.1. Installation of R package

The following code installs the R package from github sources:

```
1  # install devtools if not already installed
2  install.packages("devtools")
3  library(devtools)
4  install_github("joshchang/calonesurv")
5  library(calonesurv)
```

## B.2. Simulation of data used in this paper

We simulated draws from the populations mentioned in the main text using the following R code:

```
1   library(survival)
2   library(calonesurv)
3
4   p = 0.75
5   n = 20
6   samples = 10000
7   tvals = seq(0,1/explambda*4,by=0.05)
8
9   explambda = 1/4
10  wshape = 1
11  wscale = 5
12
13  mix_results <- matrix(nrow = length(tvals), ncol = samples)
14  pure_results <- matrix(nrow = length(tvals), ncol = samples)
15
16  for(i in 1:samples){
17    n1 = sum(rbinom(n,1,p) )
18    n2 = n-n1
19    #while(n2<2){
20    #  n1 = sum(rbinom(n,1,p) )
21    #  n2 = n-n1
22    #}
23    t1 = sort(rexp(n1,explambda))
24    t2 = sort(rweibull(n2,wshape,wscale))
25
26    km1 = cadlag(survfit(Surv(t1)~1))
27    km3 = cadlag(survfit(Surv(c(t1,t2))~1))
28    pure <- cadlag(km3$time,c(1,km3$surv))
29
30    if(n2>0){
31      km2 = cadlag(survfit(Surv(t2)~1))
32      mix <- n1/n*km1 + n2/n*km2
33      mix_results[,i] <- mix(tvals)
34    }
35    else{
36      mix_results[,i] <- pure(tvals)
37    }
38
39    pure_results[,i] <- pure(tvals)
40  }
```

## B.3. Real-world example

```
1   require(coin)
2   require(survival)
3   source("Theta.R")
4   library(data.table)
5   library(ggplot2)
6
7   surv_data = with(subset(survival::lung,ph.ecog %in% 0:2),
8                 data.frame(population = sex,
9                            censor = as.numeric(status==1),
10                           time = time, cohort = ph.ecog ))
11
12  out = Theta_hat(surv_data)
13  print(out)
14  print(confint(out))
15  print(pvalue.Theta_hat(out))
```

```
1
2    J = 10000
3    N = c(40,80,136)
4
5
6    typeone = data.table("method" = rep(0,J*length(N)*3), "n" = rep(0,J*length(N)*3), "P" = rnorm(J*length(N)*3))
7    Thetavals = data.table("n" = rep(0,J*length(N)), "Theta" = rep(0,J*length(N)))
8
9    # Simulations for evaluating Type-I error
10
11   k = 0
12   j = 1
13   for(n in N){
14     i = 1
15     while(i< J*3){
16       indices = sample(1:136,n)
17       s_data = subset(surv_data,population==1)[indices,]
18       s_data$population[1:(n/2)] = 2
19
20       tryCatch({
21         out = Theta_hat(s_data)
22         p_Theta_hat = pvalue.Theta_hat(out)
23         p_lr = pvalue(logrank_test(Surv(time,1-censor)~as.factor(population), data = s_data))
24         fit = survfit(Surv(time,1-censor)~as.factor(population), data = s_data)
25         t1 = ten(fit)
26         invisible(comp(t1))
27         p_w = attr(t1,'lrt')$pNorm[2]
28         Thetavals[j] = list( "n" =n , "Theta" = as.numeric(out))
29         typeone[J*3*k + i,] = list("method"=1,"n"=n,"P" = p_Theta_hat)
30         typeone[J*3*k + i+1,] = list("method"=2,"n"=n,"P" = p_w)
31         # logrank is 3
32         typeone[J*3*k + i+2,] = list("method"=3,"n"=n,"P" = p_lr)
33         i = i + 3
34         j = j + 1
35       },error = function(err) print(err))
36
37     }
38     k = k+1
39   }
40   pdf("fig6.pdf",family="CM Roman", width=8, height=4)
41   typeone$method = as.factor(typeone$method)
42   levels(typeone$method) = c("Our Method", "Wilcoxon signed rank","Log-rank")
43   pp = ggplot(typeone, aes(factor(n),P))
44   print(pp + geom_violin(aes(fill=factor(method)))+geom_hline(aes(yintercept=0.05),linetype='dashed')+ ylim(1e-4,1)+
45     scale_y_continuous(trans='log2',breaks = c(0.0005,0.005,0.05,0.5,1),limits = c(1e-4,1)) + theme_bw() + theme(legend.position="
                top") +
46     ylab("P-value") + xlab("Sample size") + guides(fill=guide_legend(title="Method: ")))
47   dev.off()
48
49
50   typeone$reject = typeone$P < 0.05
51   print(ftable(reject ~ method + n, data = typeone))
52
53
54   # Simulations for evaluating asymptotic convergence in distribution
55
56   Thetavals_2 = data.table("n" = rep(0,J*length(N)), "Theta" = rep(0,J*length(N)))
57
58   j = 1
59   for(n in c(50,60,70)){
60     i = 1
61     while(i< J*3){
62       indices = sample(1:136,n)
63       s_data = subset(surv_data,population==1)[indices,]
64       s_data$population[1:(n/2)] = 2
65       tryCatch({
66         out = Theta_hat(s_data)
67         Thetavals_2[j] = list( "n" =n , "Theta" = as.numeric(out))
68         j = j + 1
69         i = i + 3
70       },error = function(err) print(err))
71
72     }
73   }
74
75   # compute the theoretical Gaussian densities
76
77   n = 50
78   indices = sample(1:136,n)
79   s_data = subset(surv_data,population==1)[indices,]
80   s_data$population[1:(n/2)] = 2
81   out_50 = Theta_hat(s_data)
82
83   n = 60
84   indices = sample(1:136,n)
85   s_data = subset(surv_data,population==1)[indices,]
86   s_data$population[1:(n/2)] = 2
87   out_60 = Theta_hat(s_data)
88
89   n = 70
90   indices = sample(1:136,n)
91   s_data = subset(surv_data,population==1)[indices,]
92   s_data$population[1:(n/2)] = 2
93   out_70 = Theta_hat(s_data)
94
95   grid <- with(dd, seq(-750,750, length = 200))
96   normaldens =
97     data.frame(
98       Theta = rep(grid,3),
99       n = rep(c(50,60,70),each=length(grid)),
100      density = c(dnorm(grid, mean = 0, sd = sqrt(slot(out_50,'variance'))),
101                  dnorm(grid, 0, sqrt(slot(out_60,'variance'))),
102                  dnorm(grid, 0, sqrt(slot(out_70,'variance'))))
103    )
104
105
106  pdf("fig7.pdf",family="CM Roman", width=8, height=3)
107  pp2 = ggplot(Thetavals_2,aes(Theta)) + geom_histogram(aes(x = Theta, y = ..density..),bins = 80,fill = "grey") + facet_wrap(~n)
                 +
108    theme_bw() + xlab(expression(hat(Theta)))+
109    geom_line(aes(y = density), data = normaldens, colour = "red", size=1.1)
110  print(pp2)
111  dev.off()
```

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
