## [Reviewer comments · Royal Society Open Science]

Review History

RSOS-180496.R0 (Original submission)

Review form: Reviewer 1

Is the manuscript scientifically sound in its present form?

No

Are the interpretations and conclusions justified by the results?

No

Is the language acceptable?

No

Is it clear how to access all supporting data?

No

Do you have any ethical concerns with this paper?

No

Have you any concerns about statistical analyses in this paper?

No

Recommendation?

Major revision is needed (please make suggestions in comments)

Comments to the Author(s)

Major Concerns:

The paper proposed a cohort-weighted method to address the issue of potential heterogeneity within a population in the comparison of time-to-event curves. Sub-populations are used in the method and the simulations. But the paper doesn't describe how to determine which cohort each data point (each individual) falls into in a real-world setting, which the reviewer considers crucial. Because in order to use the proposed method after given a real-world dataset, we have to first know which cohort each individual comes from. The reviewer would ask the authors to add this missing piece to make the method complete.

Also, the reviewer is interested in a real-world data example using the proposed method.

Minor Concerns:

The simulation seems to be overly simple. The reviewer would ask the authors to explore more complicated scenarios, for example, >2 cohorts in one population, and different cohort prevalence q.

Review form: Reviewer 2 (Jiancang Zhuang)

Is the manuscript scientifically sound in its present form?

Yes

Are the interpretations and conclusions justified by the results?

Yes

Is the language acceptable?

Yes

Is it clear how to access all supporting data?

No

Do you have any ethical concerns with this paper?

No

Have you any concerns about statistical analyses in this paper?

No

Recommendation?

Major revision is needed (please make suggestions in comments)

Comments to the Author(s)

Please see attached file. (See Appendix A)

Decision letter (RSOS-180496.R0)

09-Aug-2018

Dear Dr Chang,

The editors assigned to your paper ("Empirical process-based large sample properties of the area bounded by cohort-weighted Kaplan Meier curves") have now received comments from reviewers. We would like you to revise your paper in accordance with the referee and Associate Editor suggestions which can be found below (not including confidential reports to the Editor). Please note this decision does not guarantee eventual acceptance.

Please submit a copy of your revised paper before 01-Sep-2018. Please note that the revision deadline will expire at 00.00am on this date. If we do not hear from you within this time then it will be assumed that the paper has been withdrawn. In exceptional circumstances, extensions may be possible if agreed with the Editorial Office in advance. We do not allow multiple rounds of revision so we urge you to make every effort to fully address all of the comments at this stage. If deemed necessary by the Editors, your manuscript will be sent back to one or more of the original reviewers for assessment. If the original reviewers are not available, we may invite new reviewers.

- Data accessibility

<http://datadryad.org/submit?journalID=RSOS&manu=RSOS-180496>

- Competing interests

- Authors' contributions

- Acknowledgements

- Funding statement

Please note that Royal Society Open Science charge article processing charges for all new submissions that are accepted for publication. Charges will also apply to papers transferred to Royal Society Open Science from other Royal Society Publishing journals, as well as papers submitted as part of our collaboration with the Royal Society of Chemistry (<http://rsos.royalsocietypublishing.org/chemistry>). If your manuscript is newly submitted and subsequently accepted for publication, you will be asked to pay the article processing charge, unless you request a waiver and this is approved by Royal Society Publishing. You can find out more about the charges at <http://rsos.royalsocietypublishing.org/page/charges>. Should you have any queries, please contact openscience@royalsociety.org.

on behalf of Prof. Mark Chaplain (Subject Editor)
openscience@royalsociety.org

Comments to Author:
Reviewers' Comments to Author:
Reviewer: 1

Comments to the Author(s)
Major Concerns:

The paper proposed a cohort-weighted method to address the issue of potential heterogeneity within a population in the comparison of time-to-event curves. Sub-populations are used in the method and the simulations. But the paper doesn't describe how to determine which cohort each data point (each individual) falls into in a real-world setting, which the reviewer considers crucial. Because in order to use the proposed method after given a real-world dataset, we have to first know which cohort each individual comes from. The reviewer would ask the authors to add this missing piece to make the method complete.

Also, the reviewer is interested in a real-world data example using the proposed method.

Minor Concerns:

The simulation seems to be overly simple. The reviewer would ask the authors to explore more complicated scenarios, for example, >2 cohorts in one population, and different cohort prevalence q .

Reviewer: 2

Comments to the Author(s)
Please see attached file.

Author's Response to Decision Letter for (RSOS-180496.R0)

See Appendix B.

RSOS-180496.R1 (Revision)

Review form: Reviewer 1

Is the manuscript scientifically sound in its present form?

Yes

Are the interpretations and conclusions justified by the results?

Yes

Is the language acceptable?

Yes

Is it clear how to access all supporting data?

No

Do you have any ethical concerns with this paper?

No

Have you any concerns about statistical analyses in this paper?

No

Recommendation?

Major revision is needed (please make suggestions in comments)

Comments to the Author(s)

This manuscript proposed a test statistic that was based on a cohort-weighted survival function estimator. The reviewer would like to see the following revisions:

- 1) Please add a real world example to demonstrate how the proposed test statistic can be applied.
- 2) Please add simulations to show that the Type 1 error rate has not been inflated using the proposed method.
- 3) The commonly-used method, for example, a log-rank test, can compare the hazard ratios of the two groups, on top of the P-values. But the proposed method will only provide P-values. Therefore we need to beware of the differences in interpretation.

Review form: Reviewer 2 (Jiancang Zhuang)

Is the manuscript scientifically sound in its present form?

Yes

Are the interpretations and conclusions justified by the results?

Yes

Is the language acceptable?

Yes

Is it clear how to access all supporting data?

Yes

Do you have any ethical concerns with this paper?

No

Have you any concerns about statistical analyses in this paper?

No

Recommendation?

Accept with minor revision (please list in comments)

Comments to the Author(s)

The revised manuscript is now much easier to read. The only part that needs improvement is the quality of Figures 2 and 3.

1. Figure 2. I cannot see the difference clearly. Use logarithm for x-axis?

2. Figure 3. Curves in the legend is not the same as in the plot. The differences in the plots in the first row are difficult to see.

3. In the last row of Figure 3. The estimation errors are larger for the Cohort-Weighted estimator than those for the pure KM estimator. Does this mean the Cohort-Weighted estimator is worse than the KM estimator? Do you have a large-sample example to show the advantages of the CW-KM estimator?

Decision letter (RSOS-180496.R1)

20-Sep-2018

Dear Dr Chang:

Manuscript ID RSOS-180496.R1 entitled "Asymptotic convergence in distribution of the area bounded by prevalence-weighted Kaplan-Meier curves using empirical process modeling" which you submitted to Royal Society Open Science, has been reviewed. The comments of the reviewer(s) are included at the bottom of this letter.

Please submit a copy of your revised paper before 13-Oct-2018. Please note that the revision deadline will expire at 00.00am on this date. If we do not hear from you within this time then it will be assumed that the paper has been withdrawn. In exceptional circumstances, extensions may be possible if agreed with the Editorial Office in advance. We do not allow multiple rounds of revision so we urge you to make every effort to fully address all of the comments at this stage. If deemed necessary by the Editors, your manuscript will be sent back to one or more of the original reviewers for assessment. If the original reviewers are not available we may invite new reviewers.

- Ethics statement

- Data accessibility

- Competing interests

- Authors' contributions

- Acknowledgements

- Funding statement

Please note that Royal Society Open Science charge article processing charges for all new submissions that are accepted for publication. Charges will also apply to papers transferred to Royal Society Open Science from other Royal Society Publishing journals, as well as papers submitted as part of our collaboration with the Royal Society of Chemistry (<http://rsos.royalsocietypublishing.org/chemistry>). If your manuscript is newly submitted and subsequently accepted for publication, you will be asked to pay the article processing charge, unless you request a waiver and this is approved by Royal Society Publishing. You can find out more about the charges at <http://rsos.royalsocietypublishing.org/page/charges>. Should you have any queries, please contact openscience@royalsociety.org.

on behalf of Prof. Mark Chaplain (Subject Editor)
openscience@royalsociety.org

Reviewer comments to Author:

Reviewer: 1

Comments to the Author(s)

This manuscript proposed a test statistic that was based on a cohort-weighted survival function estimator. The reviewer would like to see the following revisions:

- 1) Please add a real world example to demonstrate how the proposed test statistic can be applied.
- 2) Please add simulations to show that the Type 1 error rate has not been inflated using the proposed method.
- 3) The commonly-used method, for example, a log-rank test, can compare the hazard ratios of the two groups, on top of the P-values. But the proposed method will only provide P-values. Therefore we need to beware of the differences in interpretation.

Reviewer: 2

Comments to the Author(s)

The revised manuscript is now much easier to read. The only part that needs improvement is the quality of Figures 2 and 3.

1. Figure 2. I cannot see the difference clearly. Use logarithm for x-axis?
2. Figure 3. Curves in the legend is not the same as in the plot. The differences in the plots in the first row are difficult to see.
3. In the last row of Figure 3. The estimation errors are larger for the Cohort-Weighted estimator than those for the pure KM estimator. Does this mean the Cohort-Weighted estimator is worse

than the KM estimator? Do you have a large-sample example to show the advantages of the CW-KM estimator?

Author's Response to Decision Letter for (RSOS-180496.R1)

See Appendix C.

RSOS-180496.R2 (Revision)

Review form: Reviewer 1 (Xun Lin)

Is the manuscript scientifically sound in its present form?

Yes

Are the interpretations and conclusions justified by the results?

Yes

Is the language acceptable?

Yes

Is it clear how to access all supporting data?

Yes

Do you have any ethical concerns with this paper?

No

Have you any concerns about statistical analyses in this paper?

No

Recommendation?

Accept as is

Comments to the Author(s)

Thank the authors for addressing the comments.

Review form: Reviewer 2 (Jiancang Zhuang)

Is the manuscript scientifically sound in its present form?

Yes

Are the interpretations and conclusions justified by the results?

Yes

Is the language acceptable?

Yes

Is it clear how to access all supporting data?

Yes

Do you have any ethical concerns with this paper?

No

Have you any concerns about statistical analyses in this paper?

No

Recommendation?

Accept as is

Comments to the Author(s)

The authors have clarified my questions in the revision. I have no further suggestions except

Line 47. Figure 4 -> Figure 5.

Decision letter (RSOS-180496.R2)

17-Oct-2018

Dear Dr Chang,

I am pleased to inform you that your manuscript entitled "Asymptotic convergence in distribution of the area bounded by prevalence-weighted Kaplan-Meier curves using empirical process modeling" is now accepted for publication in Royal Society Open Science.

Kind regards,

Royal Society Open Science Editorial Office
Royal Society Open Science
openscience@royalsociety.org

on behalf of Prof. Mark Chaplain (Subject Editor)
openscience@royalsociety.org

Associate Editor Comments to Author:

With the exception of a typo spotted by one of the referees, this paper is now ready for acceptance -- congratulations! Please ensure that you tackle this minor modification during typesetting.

Reviewer comments to Author:

Reviewer: 1

Comments to the Author(s)

Thank the authors for addressing the comments.

Reviewer: 2

Comments to the Author(s)

The authors have clarified my questions in the revision. I have no further suggestions except

Line 47. Figure 4 -> Figure 5.

Appendix A

Comments on “Empirical process-based large sample properties of the area bounded by cohort-weighted Kaplan Meier curves” by A. Heuser, M. Huynh, & J. Chang

The theme of the article seems to be on the comparison between two KM estimators, $\hat{\theta}_t$ and \hat{S}_t , for cohort-level samples. I use the word “seems” because the whole article does not keep on the track of such a comparison. In the abstract, the authors state that this paper derives the properties of the Kaplan-Meier product-limit estimator when individuals are separated by cohort. But in Page 5, the authors say that the main results is related to the static defined in Equation (2.4), which is used to test whether two disjoint sets of population have the same survival functions. It is not clear that how can you use this static to compare your method and the original pure KM estimator.

Section 2

1. Line 50, Page 5. It seems that a portion of the contents is missing here.

Section 3

2. A careful description of the simulation data is necessary. Is not Population 1 too simple?
3. Figure 3. The true survival functions that are used in the simulation should be superposed
4. The analysis on the simulation data should be the numerical experiment to verify the main results. However, I cannot see in the data analysis, the asymptotic properties of $\hat{\theta}_t$ and \hat{S}_t are verified. That is to say, the theoretical variance of the estimators should be compared to the simulation results.

Appendix A.

This appendix is referred at the beginning of Section 2(a). Thus, all the notations that are not yet defined should be defined or explained. However, there are many notations in the appendix used before they are defined.

1. Line 26, Page 10. What is A_i ?
2. Line 60, Page 10. What are μ_{1,z,τ_z} and τ_z ?
3. Line 18, Page 11. What is a_z ?
4. Lines 30 to 34, Page 11. What is $\hat{S}_{z,s-}(\hat{q}_z)$? Which section is “the previous section”?

References

12 to 14. The initial letters of “kaplan” and “meier” should be capitalized.

In summary, this article should be re-organized to focus on its theme in both theory and simulation verification, and the simulation part should be extended.

Appendix B

Responses to the reviewers

August 22, 2018

We would like to thank the reviews for their careful assessment of our manuscript. We know that this manuscript is extremely technical and appreciate the feedback in making it more understandable. We have made many modifications to our manuscript in the intent that they directly address the concerns raised by the reviewers. Both reviewers raised a general point about the motivations for this manuscript. We regret that we had not made the focus of our manuscript more clear.

Here, we will describe our motivations and some changes we made to the manuscript with the additional hope that it will allay Review 1's concerns. We had two main intentions in the writing of this manuscript. First was to document the technical details of the methodology that was used by the NIH and Social Security Administration (SSA) in identifying medical conditions where applicants require fast-tracking in the adjudication of disability claims. We had alluded to this application of the work previously in the discussion but now have added the following text to the introduction section:

pg 2: These complexities arose in the identification of new disorders to incorporate into the United States Social Security Administration (SSA)'s Compassionate Allowances (CAL) initiative. The CAL initiative seeks to identify candidate medical conditions for fast-tracking in the processing of disability applications. The intent of this initiative is to prioritize applicants who are most likely to die in the time-course of usual case processing so that they may receive benefits while still living. At its inception, the CAL initiative identified conditions based on the counsel of expert opinion [15]. The SSA in collaboration with the National Institutes of Health (NIH) sought to expand the list of CAL conditions systematically, using a data-based approach. Using in-part the survival estimator described in this manuscript, the NIH identified 24 conditions for inclusion into the list of conditions [15].

The methodology used in CAL is related to that of the work of Pepe and Fleming (cf. [13,14]), where a class of weighted Kaplan-Meier statistics is introduced. Though these statistics exhibit the same limitations as in the standard Kaplan-Meier case, it should be noted that [14] introduces the stratified weighted Kaplan-Meier statistic. The statistic presented here is a priori quite similar, but instead of a weighting function, includes the empirical prevalence. In doing so, the weight is no longer independent of the event time estimate, and thus requires much different methods of proof.

In addition, we provide some more details on how the NIH/SSA used this estimator in the Discussion:

pg 7: A variant of this method was used in Rasch et al. [15] in order to classify physical disorders based on severity for the sake of prioritization of processing for disability

claims. Since the underlying survival surface is non-stationary, and the fixed observation windows create progressive censoring, that paper illustrates the utility of this statistical method. In that manuscript, the cohorts were defined based on binned application entry times and a heuristic “survival surface” was generated in order to get a single overall picture of the survivability of a given disorder. The censoring parameters τ_z varied due to the finite sampling window and the fact that more-recent cohorts are not observed for as long a time period as older cohorts, as depicted in Fig 1b. It was also expected that survival by cohort would vary due to differences in health care administration and treatment between entry cohorts. The use of the empirical prevalences (\hat{q}_z) allowed the accounting for variability in disability application volume by sufferers of given disorders, conditional on entry date.

Although the most direct and natural applications of the method that we have presented here involve discretely-indexed covariates, it is possible to use this method for continuously-indexed covariates such as time by employing the binning strategy used in Rasch et al. [15]. This approach is particularly fruitful if the sampling windows are coarse and there is clear separation between cohorts to maintain statistical independence. In this situation, it may be unreasonable to expect to construct a full continuous surface for survival. Nonetheless, a possible future extension of this method might involve replacing the sum of Eq. 2.1 with an integral and using statistical regularization tools [4] in order to infer true continuously-indexed surfaces.

From more of a mathematical standpoint, the second intent of this manuscript was to prove the asymptotic properties of our modification to the Pepe-Fleming weighted survival estimator using an empirical process model. The classical estimator assumed that the weighting proportions are known constants. Our estimator $\hat{\theta}$ assumes that \mathbf{q} is estimated from the given sample. We hope that the derivation of the large sample properties from the framework of empirical processes would be useful to a reader interested in mathematical statistics.

The question of cohort-identifiability came up in both reviewers’ critiques. In our application we are assuming that one knows the cohort of the individuals. In the SSA’s dataset, individuals were identified by their primary physical impairment and the time of application. The cohorts in this application pertained to year of application for benefits. We make this point more clear in the definition of θ_t .

Pg 3: where $S_{z,t}^{(i)}$ represents the survival function for individuals of cohort z in population i , where each individual’s cohort membership is known.

Besides the changes made in direct response to the reviewers, we have made an effort to ensure that the notation we used is consistent and have made several changes. Notably, we had originally dropped some superscripts that denoted the size of the empirical process and went instead with indices representing cohort proportions. We believe that this notation was confusing so we have made appropriate modifications.

We have also reorganized the manuscript, putting the modeling section into the appendix along with the rest of the proof so that it can be read more linearly. The appendix is now largely self-contained, except in a few places where it refers to the main quantities defined in Section 2 of the text. We have made effort in making the proof more readable and provide more high-level exposition.

Reviewer 2

The theme of the article seems to be on the comparison between two KM estimators, $\hat{\theta}_t$ and \hat{S}_t , for cohort-level samples. I use the word “seems” because the whole article does not keep on the track of such a comparison. In the abstract, the authors state that this paper derives the properties of the Kaplan-Meier product-limit estimator when individuals are separated by cohort. But in Page 5, the authors say that the main results is related to the static defined in Equation (2.4), which is used to test whether two disjoint sets of population have the same survival functions. It is not clear that how can you use this static to compare your method and the original pure KM estimator.

We regret that the motivation for our article was lost. The focus is not the comparison of these estimators as this has been done before the the literature. We provide exposition on this particular weighted estimator because we prove a property of it – the asymptotics of it’s bounding area. We have added the following text:

Pg 3: The focus of this manuscript is not the properties of this survival estimator but rather the asymptotic convergence of its bounding area and the use of such a quantity for evaluating a null hypothesis.

Section 2 1. Line 50, Page 5. It seems that a portion of the contents is missing here.

We have fixed this error

Section 3 2. A careful description of the simulation data is necessary. Is not Population 1 too simple?

The aim of our simulation is to find a simple example where a family of populations is difficult to identify on the basis of mean survival time in order to demonstrate the weighted survival estimate and the statistical power. To make this point clear we have added the text:

Pg 5: Using simulations, we explored the statistical power of the test statistic $\hat{\Theta}$ in a case where populations are difficult to distinguish based purely on mean survival time.

3. Figure 3. The true survival functions that are used in the simulation should be superposed

We have added traces of the true survival functions which now appears in blue.

4. The analysis on the simulation data should be the numerical experiment to verify the main results. However, I cannot see in the data analysis, the asymptotic properties of $\hat{\theta}_t$ and \hat{S}_t are verified. That is to say, the theoretical variance of the estimators should be compared to the simulation results.

The asymptotic properties of the product limit estimator and its weighted variant are well documented in the literature. Besides referencing the Pepe-Fleming article, we also now cite an additional source establishing convergence for the unweighted KM estimator:

Pg 3: The asymptotic convergence of the product-limit estimator and weighted variants is well established [4,15].

The asymptotic variance of the test statistic $\hat{\Theta}$ is under the assumption of the null hypothesis, being that $\theta^{(1)} = \theta^{(2)}$. In our manuscript we have proven its asymptotic variance in this restricted setting of a null hypothesis. We think that rather than demonstrating convergence through a simulation, which is inferior to a proof that we provided, the salient feature of the statistic is in its power to distinguish populations in the case where $\theta^{(1)} \neq \theta^{(2)}$. Hence, that is why we provided the comparison demonstrated in Fig. 4.

Appendix A. This appendix is referred at the beginning of Section 2(a). Thus, all the notations that are not yet defined should be defined or explained. However, there are many notations in the appendix used before they are defined.

Thank you for noting this fact. We have gone through and made sure that the appendix is notationally self-sufficient through a restructuring of the manuscript. We have also worked to make the proof easier to follow by adding more references to equations along the way.

1. Line 26, Page 10. What is A_i ?

We regret the use of A_i , the meaning of which was to denote a set corresponding to the datapoint x_i . The new notation that we now use is more clear.

Pg 10:

$$\mathcal{G}_n h = \sqrt{n} (\mathbb{P}_n - \mathbb{P}) h = \frac{1}{\sqrt{n}} \sum_{i=1}^n (h(x_i) - \mathbb{P}h),$$

2. Line 60, Page 10. What are μ_{1,z,τ_z} and τ_z ?

μ_{k,z,τ_z} referred to the population size as present in an empirical process for particles in state k of cohort z at time τ_z . The time coordinate is relative to the observation window of the cohort and each cohort may have a different time at which censorship occurs τ_z , though we allow for individual members to become censored before this time. Implicit in this notation was that we were starting with a given number n_z particles in a dormant state before $t = 0$, which we now make explicit in our new notation. We have explained this notation by including the following text:

Pg 11: let $\mu_{j,z,t}^{n_z} \geq 0$ be the number of the cohort z individuals who are in state $j \in \mathbb{Z}_4$ at time t , so that

$$\sum_{j=0}^3 \mu_{j,z,t}^{n_z} = n_z \tag{1}$$

is conserved. Also, we assume that there exists $\tau_z < \infty$ when all particles either become inactive or censored so that τ_z is the infimum time where the condition

$$n_z = \mu_{2,z,t}^{n_z} + \mu_{3,z,t}^{n_z} \quad \forall t > \tau_z \tag{2}$$

holds.

3. Line 18, Page 11. What is a_z ?

The a_z refer to fixed (known) proportion of cohort z as in the original Pepe-Fleming estimator. We modify this estimator by replacing a_z with the sample proportions \hat{q}_z , which is estimated from the sample. We have made this fact more clear in the text:

Pg 11: Assume the sample contains a **known** number $n_z = a_z n$, $a_z \in (0, 1)$, of individuals of cohort z , and let $\mu_{j,z,t}^{n_z} \geq 0$ be the number of the cohort z individuals who are in state $j \in \mathbb{Z}_4$ at time t ,

4. Lines 30 to 34, Page 11. What is $\hat{S}_{z,s-}(\hat{q}_z)$? Which section is “the previous section”?

$\hat{S}_{z,s-}(\hat{q}_z)$ refers to the Kaplan-Meier estimator for cohort z , assuming that $n_z = \hat{q}_z n$ total empirical process “particles” contributed to the computation of the KM estimator. We have modified the notation, explaining it now in the modeling section:

Pg 11: For the sample of size n_z , we denote the z -type cumulative hazard by $\Lambda_{z,t}$ and respectively define the z -type cumulative hazard and survival estimates by

$$\hat{\Lambda}_{z,t}^{n_z} = \int_0^t \frac{d\mu_{2,z,s}^{n_z}}{\mu_{1,z,s-}^{n_z}} \quad (3)$$

$$\hat{S}_{z,t}^{n_z} = \prod_{s \leq t} \left(1 - d\hat{\Lambda}_{z,s}^{n_z}\right). \quad (4)$$

Appendix C

Responses to the second-round reviewers

October 2, 2018

The authors thank the reviewers for their helpful suggestions. We have responded to all of the reviewers' concerns by making appropriate modifications to our manuscript. As a result of these changes, we feel that our manuscript is much improved. We also now provide an R implementation of the overall method, available as a package. Please see below for our itemized responses to the second round of review.

Reviewer 1

1. Please add a real world example to demonstrate how the proposed test statistic can be applied.

In the updated manuscript, we have used publicly available data within the `survival` package for R in order to demonstrate the estimators. In this example, we compared survival between males and females, using ECOG performance score to organize cohorts. In the course of doing this example application, we have decided to clean up our R code and make it public as a package so that others may easily implement this test statistic. Notably, our package contains some utility functions for arithmetic with piecewise functions in R so we hope that this code will be useful.

We reference this particular application in section 3(b), titled *Evaluating Type-I error in a real world example*. The source code for this example is presented in the appendices. In Figure 5, we provide the corresponding $\hat{\theta}_t$ estimates and in the text we note the value of the statistic, its asymptotic confidence interval, and its corresponding asymptotic P -value. For reference, we also computed P -values using the other two methods mentioned in the text. Please refer to the following text from the paper:

pg 6:

We applied the survival estimator and statistic to NCCTG Lung Cancer data [13] available within the `survival` package for R. We compared the survival between male ($n^{(1)} = 136$) and female ($n^{(2)} = 90$) cancer patients, organized by ECOG performance score ($z \in \{0, 1, 2\}$) as cohort. Using males as population 1 and females as population 2, we arrived at the test-statistic estimate: $\hat{\Theta} = -961$, with 95% asymptotic confidence interval: $(-1527, -396)$, which would support rejection ($P \approx 0.0009$) of the null hypothesis ($\hat{\theta}^{(1)} = \hat{\theta}^{(2)}$) at $\alpha = 0.05$. For reference, both the Wilcoxon ($P \approx 0.0012$) and log-rank ($P \approx 0.0015$) tests referenced in Fig. 4 also rejected the null hypothesis.

2. Please add simulations to show that the Type 1 error rate has not been inflated using the proposed method.

The type-1 error is set to a pre-defined level (we used $\alpha = 0.05$). We think this question is asking whether the $\hat{\Theta}$ statistic and its asymptotic variance imply the correct Type-I error rate – in essence, asking about the validity of the asymptotic normality of the test statistic. To validate the behavior of the statistic, we resampled the male subpopulation of the lung cancer data, comparing survival between random sets of males of defined sample sizes. In Figure 6, we looked at the rejection rate of H_0 according to the asymptotic statistics of $\hat{\Theta}$ at $\alpha = 0.05$. We note that the rejection rate is slightly elevated ($\approx 6\%$) in $n = 40$, but by $n = 80$ the rate converges to the correct value. In Figure 7, we look at the convergence in distribution for intermediate sample sizes.

3. The commonly-used method, for example, a log-rank test, can compare the hazard ratios of the two groups, on top of the P-values. But the proposed method will only provide P-values. Therefore we need to beware of the differences in interpretation

Thank you for noting this point. We have mentioned it now in the text of the revised manuscript in the Discussion, as a limitation of the utility of the method.

pg 9: We note that a strong limitation of the presented method lies in its framing in terms of null hypothesis statistical testing. The $\hat{\Theta}$ statistic only provides a P -value, as opposed to other tests such as the log-rank test which provide hazard ratios as well. As a trade-off for statistical power, one is sacrificing interpretability in the form of effect sizes.

Reviewer 2

1. Figure 2. I cannot see the difference clearly. Use logarithm for x-axis?

Thank you for this suggestion. We have amended this figure to add panels zoomed into the region where differences are observable, plotted in log scale. We have also changed the values of q_2 that we display so that differences are more noticeable.

2. Figure 3. Curves in the legend is not the same as in the plot. The differences in the plots in the first row are difficult to see.

We have modified this figure, ensuring that the legend guide for the truth is correctly labeled. The differences between survival estimates will be subtle. The first row of the figure is intended to show this fact but also to show that there do in fact exist differences when one reweights the Kaplan-Meier estimator.

3. In the last row of Figure 3. The estimation errors are larger for the Cohort-Weighted estimator than those for the pure KM estimator. Does this mean the Cohort-Weighted estimator is worse than the KM estimator? Do you have a large-sample example to show the advantages of the CW-KM estimator?

Previously, we had allowed the y-axis scale to change between the panels, so that the differences between the two estimator errors are highlighted. We think that this scaling makes it difficult to see the overall behavior of the two estimators – the mutual convergence of the mean squared error to zero. In our new manuscript, we have set y-axis limits to be consistent between the panels.

In absolute terms, the performance difference in terms of error at large sample sizes is minimal. The interesting differences occur mainly in the small-sample regime. In this regime, CW-KM does better at the tails than KM, at the expense of doing worse for earlier times. In all cases,

the variance of CW-KM is lower than that of KM. This phenomenon is a bit like the variance–bias tradeoff when looking at biased versus unbiased predictors. Please see the discussion in the updated manuscript:

Pg 9: In the second and third rows of Fig. 3, one sees that this reweighted estimator has comparable performance to the pure Kaplan-Meier estimator at large sample sizes. Asymptotically, both estimators converge to the true survival function, with variance converging to zero. At small sample sizes, there are differences. The reweighted estimator has reduced variance at the cost of larger bias, in a time-dependent manner. It also appears to have smaller variance at the cost of larger error at earlier times. This error at earlier times is mitigated by decreased error at later times (better reconstruction of tails), however, the estimator variance is lower at all times. Hence, dependent on costs, for small samples, this reweighted estimator may be preferable to the pure Kaplan-Meier estimator.